# Endothelial Ca^2+^ Signaling, Angiogenesis and Vasculogenesis: Just What It Takes to Make a Blood Vessel

**DOI:** 10.3390/ijms20163962

**Published:** 2019-08-14

**Authors:** Francesco Moccia, Sharon Negri, Mudhir Shekha, Pawan Faris, Germano Guerra

**Affiliations:** 1Laboratory of General Physiology, Department of Biology and Biotechnology “L. Spallanzani”, University of Pavia, 27100 Pavia, Italy; 2Research Centre, Salahaddin University-Erbil, Erbil 44001, Iraq; 3Department of Pathological Analysis, College of Science, Knowledge University, Erbil 074016, Iraq; 4Department of Medicine and Health Sciences “Vincenzo Tiberio”, University of Molise, 86100 Campobasso, Italy

**Keywords:** endothelial cells, endothelial colony forming cells, vascular endothelial growth factor, basic fibroblast growth factor, stromal derived factor-1α, inositol-1,4,5-trisphosphate, store-operated Ca^2+^ entry, nicotinic acid adenine dinucleotide phosphate, TRPC channels

## Abstract

It has long been known that endothelial Ca^2+^ signals drive angiogenesis by recruiting multiple Ca^2+^-sensitive decoders in response to pro-angiogenic cues, such as vascular endothelial growth factor, basic fibroblast growth factor, stromal derived factor-1α and angiopoietins. Recently, it was shown that intracellular Ca^2+^ signaling also drives vasculogenesis by stimulation proliferation, tube formation and neovessel formation in endothelial progenitor cells. Herein, we survey how growth factors, chemokines and angiogenic modulators use endothelial Ca^2+^ signaling to regulate angiogenesis and vasculogenesis. The endothelial Ca^2+^ response to pro-angiogenic cues may adopt different waveforms, ranging from Ca^2+^ transients or biphasic Ca^2+^ signals to repetitive Ca^2+^ oscillations, and is mainly driven by endogenous Ca^2+^ release through inositol-1,4,5-trisphosphate receptors and by store-operated Ca^2+^ entry through Orai1 channels. Lysosomal Ca^2+^ release through nicotinic acid adenine dinucleotide phosphate-gated two-pore channels is, however, emerging as a crucial pro-angiogenic pathway, which sustains intracellular Ca^2+^ mobilization. Understanding how endothelial Ca^2+^ signaling regulates angiogenesis and vasculogenesis could shed light on alternative strategies to induce therapeutic angiogenesis or interfere with the aberrant vascularization featuring cancer and intraocular disorders.

## 1. Introduction

The vasculature is a highly branched, tree-like, tubular network, which encompasses a system of hierarchically organized arteries, veins and interconnecting capillary beds that is optimized to provide all tissues with crucial nutrients and oxygen and remove their catabolic waste [1]. Endothelial cells line the interior surface of blood vessels, also known as tunica intima, and dictate vascular branching and morphogenesis, lumen formation, and vessel wall assembly [2]. The vasculature reaches into every organ of the vertebrate body, except the avascular cornea and the cartilage [1]. Therefore, vascular endothelium constitutes a systematically disseminated tissue that weights ≈ 1 kg in a normoweight adult, with a large proportion (>600 g) lining the capillary district, and covers a surface area of 4000–6000 m^2^. It has been estimated that, if lined end-to-end, the ≈10 trillion (10^13^) interconnected endothelial cells that form the lumen of blood vessels would wrap more than four times around the circumference of the earth [3,4]. Rather than being an inert barrier that merely regulates the exchange of solutes between circulating blood and surrounding tissues, the endothelial monolayer is critical to maintain cardiovascular homeostasis, by finely tuning a wealth of critical processes, including adjustment of vascular tone according to local energy requirements, hemostasis and coagulation, and inflammation [4,5]. In addition to providing the building blocks of the vascular transport network, endothelial cells deliver paracrine (also termed angiocrine) signals to induce growth, differentiation, or repair processes in the surrounding tissues and/or to modulate their functions [2,6]. Distinct mechanisms contribute to establish and maintain a functional vascular network in the developing embryo as well as in adult tissues [1]. Vascular development is initiated by mesoderm-derived endothelial progenitor cells (EPC), which migrate into the yolk sac and coalesce to form primitive vascular channels, a process termed vasculogenesis. Thereafter, angiogenic remodeling allows this primitive vascular labyrinth to expand into a hierarchical tree-like network of arteries, arterioles, capillary beds, venules and veins. Sprouting angiogenesis consists in the budding of neovessels from pre-existing capillaries and is driven by a decrease in local oxygen tension [1,7]. This is a complex multistep process during which a leading tip cell spearheads a new sprout by migrating outward from the parental vessel towards an angiogenic signal. Subsequently, trailing endothelial stalk cells start to proliferate, thereby supporting sprout elongation, generating the trunk of new capillaries and maintaining connection with the parental vessel. Tip cells emerging from neighbouring sprouts then anastomose to build functional vessel loops, allowing the initiation of blood flow, which also contributes to arterial-venous specification of endothelial cells [1,7]. The deposition of extracellular matrix and the recruitment of mural cells, such as pericytes and smooth muscle cells, promotes vessel maturation and stabilizes new connections. The sprouting process thus iterates until angiogenic cues cease and quiescence is re-established upon restoration of oxygen supply [1,7]. Alternately, microvascular growth can be accomplished through non-sprouting or intussusceptive angiogenesis, which consists of capillary splitting by the insertion of newly formed columns of interstitial tissue, known as pillars or posts, into the vascular lumen [8]. Although sprouting angiogenesis has long been regarded as the main mechanism responsible for vascular growth and remodeling during postnatal life, it has now been established that EPC actively contribute to neovessel formation also in the adult [9,10,11]. EPC are mobilized in circulation from stem cell niches located either in the bone marrow or in the endothelial intima of existing blood vessels to replace injured/senescent endothelial cells. In addition, EPC are massively released upon an ischemic insult, home to the hypoxic site and restore the injured vascular network by stimulating local angiogenesis in a paracrine manner or by physically engrafting within neovessels [9,10]. 

Under non-pathological conditions, the angiogenic switch is tightly controlled by a complex balance between stimulatory and inhibitory signaling molecules. When angiogenic inducers, including vascular endothelial growth factor (VEGF), basic fibroblast growth factor (bFGF), platelet derived growth factor (PDGF), stromal derived factor-1α (SDF-1α) and angiopoietin-1 (ANG-1)/Tie2, are produced in excess of anti-angiogenic factors, the balance is tipped towards neovessel formation. On the other hand, when the local concentration of anti-angiogenic factors, including thrombospondin-1, endostatin, tumstatin, vasohibin, and C-X-C motif chemokine 10 (CXCL10), overwhelms that of the stimulators, the angiogenic switch is turned off [7,12]. Insufficient vessel growth, malformation and regression contribute to numerous diseases, ranging from acute myocardial infarction, hindlimb ischemia and stroke to pre-eclampsia and neurodegeneration. Conversely, excessive/aberrant vessel growth ultimately results in tumorigenesis, intraocular disorders and inflammatory disease [7,13]. Therefore, unravelling the intracellular signaling pathways that drive vascular growth and remodeling is imperative to design efficient therapeutic strategies to treat, e.g., ischemic pathologies and cancer [7,14,15]. It has long been known that an increase in intracellular Ca^2+^ concentration ([Ca^2+^]_i_) within vascular endothelial cells plays a crucial role in angiogenesis [16,17,18]. Likewise, recent work demonstrated that intracellular Ca^2+^ signals also drive vasculogenesis by stimulating EPC to undergo proliferation, migration, and tube formation both in vitro and in vivo [19,20]. Herein, we discuss the basic mechanisms of pro-angiogenic Ca^2+^ signaling in vascular endothelial cells and circulating EPC. First, we survey the distinct Ca^2+^ signatures evoked by growth factors and chemokines in endothelial cells and the Ca^2+^-dependent decoders that translate endothelial Ca^2+^ waves into a pro-angiogenic response. We will briefly mention specific members of the Transient Receptor Potential (TRP) Canonical (TRPC) sub-family, as the role of TRP channels in angiogenesis has been extensively covered in two recent, comprehensive review articles [21,22]. Then, we describe the Ca^2+^ signaling toolkit recruited by VEGF, insulin-like growth factor 2 (IGF2) and SDF-1α in circulating EPC. The majority of the studies reviewed in the present article deal with endothelial Ca^2+^ signals arising under non-pathological conditions both in vitro and in vivo. Remodeling of the Ca^2+^ handling machinery by cardiovascular and oncological diseases is a less known facet of endothelial physiology and is addressed where appropriate.

## 2. Growth Factors and Chemokines Induce Pro-Angiogenic Ca^2+^ Signals in Vascular Endothelial Cells

An increase in [Ca^2+^]_i_ has long been recognized as a key pro-angiogenic pathway that, lying at the intersection of multiple signaling cascades, is recruited by distinct mitogens to promote and modulate endothelial cell fate [23,24]. Growth factors and chemokines induce the angiogenic switch through an increase in [Ca^2+^]_i_ that stimulates endothelial cell proliferation, adhesion, migration and bidimensional tube formation [16,25,26,27]. Furthermore, laminar shear stress that arises as a result of increased collateral blood flow during arteriogenesis, may also induce pro-angiogenic Ca^2+^ signals to favor vascular remodeling [28,29]. In addition, vascular endothelial cells may receive Ca^2+^-related pro-angiogenic inputs also by vasoactive and inflammatory mediators, including thrombin [30,31], ATP [32,33,34], ADP [34], and acetylcholine [35,36], and pleiotropic hormones, such as erythropoietin [37,38]. Finally, mechanical injury of the vascular intima may induce intracellular Ca^2+^ waves both at the edge of the injured area and at more remote sites, which are likely to promote vascular repair by inducing endothelial cell proliferation and migration [39,40]. Distinct intracellular Ca^2+^ signatures have been detected depending on the nature and strength of the extracellular stimulus, on the vascular bed and the species, and on the components of the multifaceted endothelial Ca^2+^ toolkit recruited by pro-angiogenic cues [18,25,41].

### 2.1. The Endothelial Ca^2+^ Toolkit Recruited by Growth Factors and Chemokines to Stimulate Angiogenesis

Mammalian cells, including vascular endothelial cells, impinge on two sources to generate the Ca^2+^ response to extracellular stimuli (Figure 1): endogenous Ca^2+^ mobilization and extracellular Ca^2+^ entry [18,42]. The recovery of [Ca^2+^]_i_ to pre-stimulation levels is driven by a sophisticated network of Ca^2+^ pumps and transporters, such as Sarco-Endoplasmic Reticulum Ca^2+^-ATPase (SERCA), which sequesters cytosolic Ca^2+^ into the endoplasmic reticulum (ER), Plasma Membrane Ca^2+^-ATPase (PMCA) and Na^+^/Ca^2+^ exchanger (NCX), which clear Ca^2+^ across the plasma membrane (Figure 1) [43,44,45,46,47,48,49,50]. In addition, mitochondria were shown to buffer the influx of Ca^2+^ through store-operated channels, thereby redirecting entering Ca^2+^ to the ER via the mitochondrial NCX in absence of the Ca^2+^-releasing second messenger inositol 1,4,5-trisphosphate (InsP_3_) [51,52,53].

#### 2.1.1. The Onset of Pro-Angiogenic Ca^2+^ Signals: PLCβ and PLCγ

Pro-angiogenic cues bind to specific receptor tyrosine kinases (RTK) and G_q/11_-protein coupled receptors (G_q/11_PCR) which, respectively, recognize growth factors and a wide assortment of chemokines, autacoids and hormones [41,54,55]. RTK and G_q/11_PCR, in turn, trigger an increase in endothelial [Ca^2+^]_i_ by recruiting, respectively, phospholipase C-γ (PLCγ1-2) and phospholipase C-β (PLCβ1-4) (Figure 1). Although both PLCγ1 and PLCγ2 are present in vascular endothelial cells [56,57], PLCγ1 is regarded as the main transducer of RTK activity [58,59] as PLCγ2 role is hitherto restricted to blood lineage cells [60]. PLCβ1-4 are also readily detectable throughout vascular endothelium [56], albeit PLCβ1 is absent in human umbilical vein endothelial cells (HUVEC) [57], which represent one of the most widespread models to investigate endothelial Ca^2+^ signals [18]. It has been reported that Gα_q_ monomers activate PLCβ isotypes according to the following rank order: PLCβ1 ≥ PLCβ3 > PLCβ2, while PLCβ4 recruitment is heavily limited by ribonucleotides, such as GTP-γ-S [60,61]. The role played by the distinct PLCβ isotypes in the endothelial Ca^2+^ response to G_q/11_PCR has not been carefully dissected, but preliminary evidence suggested the involvement of PLCβ1 [62] and PLCβ3 [63]. In addition, PLCβ1-3, but not PLCβ4, are also sensitive to Gβγ dimers, although they display high affinity only towards PLCβ2 [60]. As a consequence, pro-angiogenic Ca^2+^ signals may also be induced by G_i/o_PCR, such as P2Y_12_ [64,65], sphingosine-1 phosphate (S1P) receptor 1 (S1R1) [66], and C-X-C chemokine receptor type 4 (CXCR4) [67].

Once engaged by extracellular stimulation, PLCβ and PLCγ isoenzymes catalyze the hydrolysis of phosphatidylinositol 4,5-bisphosphate (PIP_2_), a minor membrane phospholipid, into the two intracellular second messengers, InsP_3_ and diacylglycerol (DAG) (Figure 1) [42]. As mentioned above, PLCγ1 is the main PLC isozyme whereby growth factors, such as VEGF, the master regulator of angiogenesis, regulate endothelial cell proliferation, migration, and tube formation [58,59,68]. For instance, targeted next generation sequence recently identified a PLCγ1 with a recurrent nonsynonymous mutation (R707Q) in primary cardiac angiosarcomas [69], a rare set of tumors triggered by aberrant proliferation and migration of coronary endothelial cells. Expression studies revealed that the PLCγ1-R707Q mutant was constitutively active in HUVEC, thereby enhancing InsP_3_ synthesis and the Ca^2+^-dependent activation of calcineurin and increasing endothelial cell migration and invasiveness [69]. Surprisingly, a recent series of studies demonstrated that VEGF-induced InsP_3_ production and downstream pro-angiogenic effects are also mediated by PLCβ3 [70,71]. Furthermore, genetic silencing of PLCβ3 impaired proliferation, migration and tubulogenesis in HUVEC grown in the presence of EGM-2 [72], an endothelial growth medium enriched with multiple growth factors. It has, indeed, been shown that the RTK VEGF receptor 2 (VEGFR2), which is the main signaling VEGF receptor in vascular endothelial cells [24], may recruit PLCβ3 through phosphorylation at serine 537 and 1105 (S537 and S1105) [70]. PLC activation leads to a pro-angiogenic increase in [Ca^2+^]_i_ that can be initiated by both InsP_3_ [58,68,69,70,73] and DAG [74,75,76,77,78]. In the following section, we illustrate how InsP_3_ production results in pro-angiogenic Ca^2+^ signals in vascular endothelial cells, whereas the mechanism of action of DAG will be briefly described in Section 2.1.3. 

#### 2.1.2. Endogenous Ca^2+^ Release Induced by Pro-Angiogenic Cues: InsP_3_ Receptors (InsP_3_R), Ryanodine Receptors (RyR) and Two-Pore Channels (TPC)

The ER represents the largest endogenous Ca^2+^ store in vascular endothelial cells by containing approximately 75% of the intracellular Ca^2+^ reservoir (Figure 1) [79]. The ER Ca^2+^ concentration ([Ca^2+^]_ER_) is maintained at around 100-500 μM by SERCA2b [43,47,49,50,80], an ER-specific pump displaying a high affinity for Ca^2+^, but low Ca^2+^ transport capacity [18]. Vascular endothelial cells also express SERCA3 [81,82], however, its expression decreases during proliferation in culture [81]. In addition, Ca^2+^ sequestration in endothelial ER vesicles is impaired by genetic silencing of SERCA2b, but not SERCA3 [48]. 

##### 2.1.2.1. InsP_3_R

InsP_3_ evokes endogenous Ca^2+^ release by binding to the non-selective cation channels, InsP_3_R [18,83,84], which are located in the ER membrane (Figure 1). The resulting opening of InsP_3_R leads to the efflux of intraluminal Ca^2+^ along the electrochemical gradient between the ER and the cytosol, where [Ca^2+^]_i_ quickly raises from around 100 nM up to 1 μM [18,83]. InsP_3_R modulate a wealth of processes, including proliferation, migration, gene expression, fluid secretion, synaptic plasticity, contraction, and membrane excitability [85]. Three InsP_3_R isoforms, i.e., InsP_3_R1-3, have been described in mammalian cells [86], all of which are present throughout vascular endothelium [18], although their pattern of expression may vary depending on the species and the vascular bed [81,83,87,88,89,90,91]. We refer the readers to a recent review, which provides a comprehensive overview of endothelial InsP_3_R [83]. Herein, we just recall that InsP_3_R are actually primed by InsP_3_ to respond to stimulation to ambient Ca^2+^, which gates InsP_3_R at low concentrations (≈50–200 nM), but inhibits their opening at substantially higher levels [86]. The bell-shaped dependence of InsP_3_R on surrounding Ca^2+^ has been also demonstrated in vascular endothelial cells [92]. The InsP_3_R subtypes differ in their affinity for InsP_3_ (InsP_3_R2 > InsP_3_R1 > InsP_3_R3) and in their sensitivity to Ca^2+^-induced inhibition, as InsP_3_R3 is less prone to close in the presence of high Ca^2+^ concentration [93]. As a consequence of their peculiar sensitivity to InsP_3_ and Ca^2+^, InsP_3_R1 and InsP_3_R2 are more tailored to generate long-lasting oscillations in [Ca^2+^]_i_, while InsP_3_R3 rather instigates monophasic Ca^2+^ signals [83,93]. No study has hitherto evaluated the correlation between the pattern of InsP_3_R endowed to a given endothelial cell and the induction of intracellular Ca^2+^ oscillations by pro-angiogenic cues. However, two recent studies demonstrated that acetylcholine induces repetitive Ca^2+^ transients in mouse brain endothelial cells, which express both InsP_3_R1 and InsP_3_R2 [89], but a monophasic increase in [Ca^2+^]_i_ in human cerebrovascular endothelial cells, in which InsP_3_R3 is the most abundant isoform [88]. It is, finally, worth recalling here that early work provided the evidence that InsP_3_Rs may be located also on the plasma membrane, and therefore mediate extracellular Ca^2+^ entry, in bovine aortic endothelial cells (BAEC) [94] and HUVEC [95]. This observation has been subsequently confirmed in other cell types and increases the versatility of InsP_3_ signaling in mammalian cells [96].

##### 2.1.2.2. RyR

InsP_3_-induced intracellular Ca^2+^ release may be amplified by the recruitment of adjoining RyR through the process of Ca^2+^-induced Ca^2+^ release [18]. RyR mainly modulate muscle contraction, although they have also been involved in synaptic transmission in the brain, insulin release from pancreatic β-cells and control of vascular tone in endothelial cells [97]. Of the three RyR isoforms, i.e., RyR1-3, expressed in mammalian cells [42], vascular endothelial cells mainly express RyR3 [98]. It should, however, be pointed out that several studies failed to detect the endothelial expression of RyR [87,88,89] and that single-cell RT-PCR revealed that RyR3 was present in a limited percentage (5–25%) of the analyzed endothelial cells [98]. Consistently, although RyR-mediated endogenous Ca^2+^ release has been reported [99,100,101], the role of RyR in endothelial Ca^2+^ signaling is usually regarded as inconsistent or not as necessary as that played by InsP_3_R [39,84,102,103]. However, a recent investigation suggested that RyR may contribute to VEGF-induced Ca^2+^ signals in human aortic endothelial cells (HAEC) [48].

##### 2.1.2.3. TPC

While the ER is the largest endogenous Ca^2+^ store recruited by endothelial mitogens, emerging evidence hinted at the acidic vesicles of the endolysosomal (EL) system as an alternative Ca^2+^ reservoir establishing a Ca^2+^-mediated cross-talk with the ER to generate pro-angiogenic outputs [41,104]. The EL Ca^2+^ store is targeted by the latest addition to the family of Ca^2+^-releasing messengers, namely nicotinic acid adenine dinucleotide phosphate (NAADP), that release EL Ca^2+^ by activating TPC 1 and 2 (TPC1-2) (Figure 1), which are novel members of the superfamily of voltage-gated Ca^2+^ channels [105]. NAADP and TPC1-2 regulate a growing number of functions, including autophagy, nutrient sensing, membrane trafficking, exocytosis, fertilization and embryogenesis, proliferation and synaptic transmission [105,106,107,108]. NAADP synthesis has been attributed to the multifunctional enzyme CD38, although NAADP levels are increased, rather than decreased, by genetic suppression of CD38 in several murine tissues [109]. However, CD38 is expressed in endothelial cells [110] and NAADP is synthesized in response to endothelial autacoids, such as histamine [111]. In addition, NAADP supports the angiogenic Ca^2+^ response to VEGF in HUVEC [104]. It has been proposed that NAADP induces local EL Ca^2+^ release through TPC1-2 that is, in turn, amplified into a regenerative Ca^2+^ wave by the Ca^2+^-dependent recruitment of juxtaposed (<30 nm) InsP_3_R and RyR at quasi-synaptic ER-EL junctions [112,113]. The so-called “trigger hypothesis”, a term introduced by Antony Galione [112], has been invoked to describe the role played by NAADP in endothelial Ca^2+^ signaling [88,104,111,114]. However, this Ca^2+^-mediated cross-talk is bidirectional, as InsP_3_R at the ER-EL interface may in turn refill acidic vesicles with Ca^2+^, thereby modulating the EL Ca^2+^ content [115,116]. Whether this reciprocal EL-to-ER Ca^2+^ shuttle exists and plays an angiogenic role in vascular endothelial cells is still unclear.

#### 2.1.3. Extracellular Ca^2+^ Entry Induced by Pro-Angiogenic Cues in Vascular Endothelial Cells: Store-Operated Ca^2+^ Entry (SOCE), STIM1 and Orai1, and TRPC

SOCE is the most widespread Ca^2+^ entry pathway in both excitable and non-excitable cells [117,118] and regulates a plethora of cellular functions, including refilling of the ER Ca^2+^ store, gene expression, cell cycle regulation, cytoskeletal remodeling, NO release and cyclic AMP production [117,119]. SOCE also represents the main Ca^2+^-permeable pathway, which mediates extracellular Ca^2+^ entry in response to external autacoids and angiogenic cues in vascular endothelial cells [15,120,121]. Endothelial SOCE is activated upon depletion of the InsP_3_-sensitive ER Ca^2+^ pool and may, therefore, be activated by RTK, such as VEGFR2 [48,122,123,124]. The molecular mechanisms underlying endothelial SOCE vary depending on the species, the vascular district and the activating stimulus, as recently reviewed in [15,22,120,121]. Therefore, herein we mainly refer to SOCE activation by pro-angiogenic cues in vascular endothelial cells (Figure 1). 

##### 2.1.3.1. SOCE: STIM1 and Orai1

Early work carried out by Mohamed Trebak and coworkers revealed that genetic silencing of STIM1 and Orai1 abolish SOCE in HUVEC (Figure 1) [122]. This finding was later confirmed by at least other three independent studies [123,125,126]. STIM1 provides the ER Ca^2+^ sensor, which detects ER Ca^2+^ depletion in response to InsP_3_-synthesizing stimuli, such as growth factors and chemokines. Then, STIM1 undergoes a complex conformational remodeling which results in the relocation of STIM1 oligomers within the peripheral cisternae of the ER. Herein, STIM1 oligomers assembly into localized clusters, known as puncta, which are ≈20 nm apart from the plasma membrane, thereby tethering and gating Orai1, which constitutes the pore-forming subunit of store-operated channels [15,122,123]. Orai1 channels are hexameric, bind to STIM1 proteins in a 1:2 ratio, and mediate a highly Ca^2+^-permeable channel, known as Ca^2+^-release activated Ca^2+^ current (I_CRAC_) [117]. The I_CRAC_ exhibits distinguishing biophysical features, including extremely low unitary conductance (10–25 fS for Ca^2+^), inwardly-rectifying current-to-voltage (I-V) relationship, very positive reversal potential (E_rev_ ≈ +60 mV), and high selectivity for Ca^2+^ over monovalent cations (P_Ca_/P_Na_ > 1.000) [117]. An I_CRAC_-like current has been recorded in vascular endothelial cells in response to massive ER Ca^2+^ depletion [122,127,128] and, as expected, it is inhibited upon genetic deletion of STIM1 and Orai1 [122]. Nevertheless, VEGF-evoked I_CRAC_ is too tiny to be detected by conventional recording systems (i.e., it falls below the pA range) and is yet to be measured [123]. 

##### 2.1.3.2. SOCE: STIM2 and Orai2-3

In addition to STIM1 and Orai1, vascular endothelial cells express their paralogues, STIM2 and Ora1-3 [88,89,122,123,129,130]. Endothelial STIM2 regulates basal Ca^2+^ entry and ER Ca^2+^ loading by regulating resting Ca^2+^ permeability [131], whereas it is unknown whether it also controls the I_CRAC_ arising in response to physiological stimulation, as shown elsewhere [132,133]. The sole exception is currently provided by a human brain microvascular endothelial cell line, hCMEC/D3, in which STIM2 is the only STIM isoform expressed [88]. Interestingly, STIM2 deletion impairs proliferation in HUVEC although its involvement in SOCE was not investigated [122]. Orai2, in turn, is the most likely candidate to mediate SOCE in the mouse cerebrovascular endothelial cell line, bEND5, in which Orai1 and Orai3 are absent [89]. In addition, Orai2 could serve as a negative modulator of Orai1 and SOCE in a bovine brain capillary endothelial cell line, t-BBEC117, in which Orai2 up-regulation during the G2/M phase may retard the proliferation rate [129]. Finally, the endothelial Orai3 is not sensitive to STIM proteins, but to the arachidonic acid-derived metabolite, leukotriene C4 [134].

##### 2.1.3.3. SOCE: TRPC1 and TRPC4

The molecular make-up of the endothelial SOCE is, however, a matter of intense controversy [22,120,121,135]. It has long been known that multiple members of the TRPC sub-family of non-selective cation channels may also support SOCE in response to pharmacological, i.e., upon SERCA inhibition, and physiological, i.e., by InsP_3_-synthesizing autacoids, stimuli [21,22]. The TRPC sub-family encompasses seven members (TRPC1-TRPC7) that, with the notable exception of the pseudogene TRPC2 in humans, enable Na^+^ and Ca^2+^ entry downstream of PLC activation [22,136]. TRPC channels control a multitude of functions, including gene expression, cell motility and adhesion, membrane excitability, angiogenesis and NO release [136,137,138]. Classically, endothelial TRPC3 and TRPC6 mediate store-independent Ca^2+^ influx following PLCγ1 activation [21,22]. For instance, TRPC3 and TRPC6 serve as second messengers-operated Ca^2+^-permeable channels directly gated by DAG (Figure 1) [139]. Intriguingly, TRPC1 and TRPC4 were shown to interact with STIM1 and mediate SOCE in human and mouse lung endothelial cells [140], whereas SOCE was severely down-regulated in cultured aortic endothelial cells isolated from TRPC4-deficient mice [141]. Likewise, a role for endothelial SOCE has been proposed for TRPC1 [124,142] and TRPC4 [143,144] by other independent studies (Figure 1). TRPC channels show a distinct repertoire of biophysical properties as compared to Orai1, which result in a store-operated current, also termed I_SOC_, which is featured by: 1000 times higher unitary conductance (in the pS range), linear I-V relationship and E_rev_ of ≈0 mV as TRPC channels do not discriminate between Na^+^, K^+^ and Ca^2+^ [145,146]. Quite surprisingly, however, electrophysiological recordings revealed that the endothelial TRPC1 and TRPC4 channels mediate either an I_CRAC_-like current [141] or a less Ca^2+^-selective current with biophysical properties intermediate between I_CRAC_ and I_SOC_ [142,143,147,148]. Alternately, endothelial TRPC channels may line the pore of an I_SOC_-like conductance [124,149,150,151]. A recent series of studies sought to solve this conundrum by suggesting that one TRPC1 and two TRPC4 subunits assemble into a supermolecular ternary complex in vascular endothelial cells [152,153]. The sensitivity of this TRPC1/TRPC4 complex to ER Ca^2+^ loading is conferred by STIM1 [154], which could interact with both TRPC1 and TRPC4 [152], and by the interaction between TRPC4 and protein 4.1 and between protein 4.1 and spectrin [142,143]. In addition, Orai1 may constitutively interact with TRPC4, and potentially with TRPC1 upon ER Ca^2+^ depletion, thereby increasing the Ca^2+^-selectivity and conferring the I_CRAC_-like fingerprints to the endothelial store-operated current [153,155]. While this information fulfills the goal to complete our description of the mechanisms that underpin the endothelial SOCE, only limited evidence has been provided to support the role of store-operated TRPC1 and TRPC4 in the Ca^2+^ response to pro-angiogenic cues.

## 3. Pro-Angiogenic Ca^2+^ Signals in Vascular Endothelial Cells

The Ca^2+^ toolkit described in Paragraph 2 may be differently recruited by distinct growth factors and cytokines to induce spatio-temporal Ca^2+^ signals, which are precisely tailored to regulate specific phases of the angiogenic process. The endothelial Ca^2+^ response to pro-angiogenic cues usually consists of a transient or biphasic increase in [Ca^2+^]_i_ or in repetitive Ca^2+^ spikes. These Ca^2+^ signatures are mainly driven by endogenous Ca^2+^ release and SOCE, although the contribution of TRPC channels has been reported. The pro-angiogenic Ca^2+^ signal can be modulated by concomitant changes in the endothelial membrane potential (V_M_) due to the recruitment of Ca^2+^-dependent conductances. Novel high-resolution and high-speed imaging techniques confirmed that VEGF-induced endothelial Ca^2+^ signals arise in vivo and drive endothelial cell sprouting and migration. 

### 3.1. VEGF-Induced Intracellular Ca^2+^ Signals in Vascular Endothelial Cells

The VEGF superfamily of growth factors include VEGF-A (which comprises four isoforms: VEGF-A_121_, VEGF-A_145_, VEGF-A_165_, VEGF-A_189_), VEGF-B, VEGF-C, VEGF-D, VEGF-E (encoded by Orf virus), VEGF-F (Vammin, isolated from *Vipera ammodytes* venom), and placenta growth factor (PlGF) [24,156]. While VEGF-C and VEGF-D mainly promote development of lymphatic vessels, VEGF-A_165_ (commonly termed VEGF) is the master regulator of angiogenesis in peripheral circulation as well as in most pathologies associated to aberrant vascular growth, such as cancer and blinding eye disorders [24,156]. VEGF isoforms stimulate vascular and lymphatic endothelial cells by binding to their high affinity cognate receptors, which include the RTK VEGFR1, VEGFR2, and VEGFR3 and the VEGF co-receptors neuropilin 1 and 2 (NRP1 and NRP2, respectively) and heparin sulfate proteoglycans. VEGFR1 and VEGFR2 are mainly expressed in vascular endothelium, while VEGFR3 is restricted to lymphatic endothelial cells. VEGFR2 [also known as KDR (kinase insert domain receptor, human) and Flk1 (fetal liver kinase-1, mouse)] is the main receptor isoform which transduces VEGF signaling in vascular endothelial cells, while VEGFR1 (also termed Fms-like tyrosine kinase 1, Flt1) may exist in a soluble form (sFlt1) which presents a higher affinity for VEGF than VEGFR2 and is, therefore, able to inhibit angiogenesis [24,156]. When VEGF binds to VEGFR2, the receptor undergoes dimerization and auto- or trans-phosphorylation of tyrosine residues on the receptor dimer as well as on downstream mediators of the pro-angiogenic signal. These include PLCγ1 and the RAS/RAF/extracellular signal-regulated kinases (ERK)/mitogen-activated protein kinase (MAPK) pathway, which promotes vascular development and arteriogenesis; the phosphoinositide 3-kinases (PI3K)/AKT pathway, which supports endothelial cell survival and limits apoptosis; endothelial nitric oxide (NO) synthase (eNOS), which stimulates endothelial cell proliferation and migration and drives the increase in capillary permeability; and SRC and small GTPases, which regulate endothelial junctions and endothelial permeability and regulate endothelial cell shape, cell migration and polarization [24,135,157,158].

An increase in [Ca^2+^]_i_ is regarded as a crucial signal whereby VEGF stimulates vascular endothelial cells to undergo cell fate specification, proliferation, migration, tubulogenesis and neovessel formation [21,24]. The first evidence about the pro-angiogenic role of endothelial Ca^2+^ signaling dates back to thirty years ago, when Criscuolo and coworkers demonstrated that the tumor-secreted vascular permeability factor, subsequently identified as VEGF by Napoleone Ferrara [156], caused a biphasic increase in [Ca^2+^]_i_ in several types of endothelial cells, including HUVEC [159]. A subsequent study revealed that the endothelial Ca^2+^ response to VEGF was mediated by VEGFR2 [160]. The majority of the work elucidating the relationship between VEGF, Ca^2+^ signaling and angiogenesis has been carried out in HUVEC. In the next chapters, therefore, we first illustrate the mechanisms whereby VEGF induces pro-angiogenic Ca^2+^ signals in HUVEC and then focus our attention on other endothelial cells types.

### 3.2. VEGF-Induced Intracellular Ca^2+^ Signals in HUVEC

The typical Ca^2+^ response to VEGF in HUVEC consists in a biphasic elevation in [Ca^2+^]_i_ as originally reported in [73] and subsequently confirmed in [122,123,161,162]. This pattern of signaling comprises an initial Ca^2+^ peak, which is due to InsP_3_-dependent Ca^2+^ release from the ER, followed by a prolonged plateau phase, which is maintained by the interaction between STIM1 and Orai1, i.e., by SOCE activation (Figure 1 and Table 1) [73,122,123]. Notably, genetic deletion (through a small interfering RNA) and pharmacological blockade (with carboxyamidotriazole and S66) of Orai1 prevents HUVEC proliferation, migration and tube formation [73,123]. In addition, VEGFR2 and Orai1 are clustered at restricted sites within the plasma membrane, a mechanism that could remarkably improve the efficiency of VEGF signaling in these cells [123]. It has also been proposed that plasma membrane InsP_3_R contribute to VEGF-induced Ca^2+^ entry in HUVEC, but the evidence in favor of this hypothesis is only correlative [95]. Conversely, strong evidence suggest that VEGF-induced extracellular Ca^2+^ entry in HUVEC may be sustained by the store-independent channels (Figure 1 and Table 1), TRPC3 [77,163] and TRPC6 [76]. TRPC3, in turn, could recruit the reverse (Ca^2+^-entry mode) of NCX1 [163]. Finally, TRPC1 could also contribute to VEGF-induced Ca^2+^ plateau in HUVEC [124,164]. TRPC1 is physically retained in close associated with VEGFR2 by Klotho protein, but how VEGFR2 activation results in TRPC1-mediated Ca^2+^ entry is unclear [164]. 

More recently, it was found that NAADP supports VEGF-induced InsP_3_-dependent Ca^2+^ release in HUVEC (Figure 1 and Table 1) [104]. Accordingly, genetic deletion of TPC1 reduces VEGF-induced intracellular Ca^2+^ signaling, in vitro tubulogenesis and neovessel formation in vivo [104]. As mentioned in Section 2.1.2.3, NAADP-dependent Ca^2+^ release from acidic stores could provide the local pulse of Ca^2+^ necessary for InsP_3_R activation by InsP_3_, but this model remains to be confirmed in HUVEC. A recent investigation compared the Ca^2+^ responses to VEGF-A_165_ and VEGF-A_121_ in HUVEC and found that their Ca^2+^ sensitivity to VEGF-A_165_ is remarkably higher due to the more efficient recruitment of PLCγ1 [161]. In addition, when VEGFR2 availability is compromised, as observed upon genetic silencing of the dual specificity tyrosine phosphorylation-regulated kinase A (DYRKA), VEGF-induced InsP_3_-dependent ER Ca^2+^ release and the subsequent engagement of Ca^2+^-sensitive decoders is hampered [58]. VEGF-induced intracellular Ca^2+^ signals have also been recorded in the HUVEC-derived cell line, EA.hy926, in which VEGF triggers a transient increase in [Ca^2+^]_i_ [165]. This Ca^2+^ signal is mediated by InsP_3_R (Table 1) and requires the concomitant production of the gasotransmitter hydrogen sulphide (H_2_S). Accordingly, the pharmacological blockade of H_2_S synthesis inhibits the Ca^2+^ response to VEGF as well as VEGF-induced EA.hy926 cell proliferation and migration [165]. H_2_S is a known modulator of the endothelial Ca^2+^ toolkit [166,167,168] and regulates angiogenesis [169], but the mechanistic link between VEGF, H_2_S and endothelial Ca^2+^ signaling deserves further investigation.

### 3.3. VEGF-Induced Intracellular Ca^2+^ Signals in Other Vascular Endothelial Cell Types: In Vitro and In Vivo Evidences

Apart from HUVEC, VEGF-induced intracellular Ca^2+^ signals have been widely characterized in peripheral and pulmonary circulation. For instance, endothelial Ca^2+^ signaling drives VEGF-induced retinal angiogenesis, whose dysregulation could lead to several sight-threatening diseases, including diabetic retinopathy, age-related macular degeneration, and retinopathy of prematurity [170]. VEGF triggers a biphasic increase in [Ca^2+^]_i_ in bovine retinal endothelial cells (BREC). As reported in HUVEC, the initial Ca^2+^ peak is patterned by InsP_3_-mediated ER Ca^2+^ release, whereas the plateau phase depends on Ca^2+^ entry through a yet to be defined Ca^2+^-permeable route. The pharmacological inhibition of InsP_3_-dependent Ca^2+^ mobilization prevents VEGF-induced BREC proliferation, migration, tubulogenesis and sprout formation [170]. Of note, genetic deletion of TRPC4 impairs VEGF-induced migration and tube formation in human retinal microvascular endothelial cells (HRMEC), but it is not clear whether TRPC4 is sustained in a store-dependent manner [171]. VEGF induces a biphasic Ca^2+^ signal also in the intact endothelium of ovine uterine vasculature [172], although it is not clear whether this Ca^2+^ response drives VEGF-induced angiogenesis and NO-dependent vasodilation during pregnancy [173]. Surprisingly, pre-treatment with VEGF inhibits ATP-induced intracellular Ca^2+^ burst and NO production [172,174], thereby mimicking the reduction in uterine vascular resistance that causes pre-eclampsia. An increase in local levels of VEGF has indeed been reported in pre-eclampsia and could be responsible for the local hypertension that presents significant risks of death for both mother and child [175]. VEGF-induced intracellular Ca^2+^ signals were also observed in mouse aortic endothelial cells (MAEC), in which the Ca^2+^ response is abolished by genetic deletion of the Ca^2+^-dependent tyrosine kinase Pyk2 [59]. Likewise, VEGF-induced migration, in vitro tubulogenesis and actin cytoskeleton remodeling are attenuated in MEAC deficient of Pyk2 [59]. An InsP_3_-dependent pro-angiogenic Ca^2+^ response to VEGF was also recorded in human dermal microvascular endothelial cells (HDMEC) [176], HAEC [177], human pulmonary artery endothelial cells [178], bovine choroidal endothelial cells [179], porcine aorta endothelial cells (PAEC) [25,180] and coronary venules endothelial cells [181]. VEGF-induced InsP_3_-dependent Ca^2+^ release is further supported by RyR and Orai1-dependent extracellular Ca^2+^ entry in HAEC (Table 1) [48]. VEGF has also been shown to induce endogenous Ca^2+^ release and extracellular Ca^2+^ entry, presumably through Orai1, in MAEC [50]. In addition, it has been suggested that VEGF underlies the spontaneous Ca^2+^ oscillations driving in vitro tubulogenesis in mouse yolk sac endothelial cells [182]. Finally, a heteromeric channel, which is likely to comprise TRPC3 and TRPC6 subunits, could mediate VEGF-induced extracellular Ca^2+^ entry in HDMEC [74,75]. 

A sophisticated tool to investigate how the endothelial Ca^2+^ toolkit shapes the pro-angiogenic Ca^2+^ response to VEGF is provided by computational modeling. A rule-based modeling approach utilizing the programming language BioNetGent recently confirmed that the extent of VEGF-induced Ca^2+^ entry during the plateau phase is finely tuned by the I_CRAC_. Likewise, a reduction in the rate of Ca^2+^ clearing through either SERCA or PMCA could further sustain the plateau amplitude [183].

Three recent studies have shed novel light on the mechanism whereby VEGF-induced endothelial Ca^2+^ signals drive angiogenesis. An investigation carried out on PAEC revealed that, as long as VEGF concentration is increased from the low to high nanomolar range, the percentage of cells displaying transient or repetitive Ca^2+^ spikes decreases, while the fraction of cells showing a biphasic Ca^2+^ signal increases [25]. As described in more detail in Section 4.3 and Section 4.5, intracellular Ca^2+^ oscillations selectively drive proliferation, while the persistent plateau phase is crucial for migration [25]. In addition, high-speed, three-dimensional (3D) time-lapse imaging disclosed repetitive intracellular Ca^2+^ waves induced by VEGF in both stalk and tip cells sprouting from dorsal aorta and posterior cardinal vein in zebrafish [184]. VEGFR2 and VEGFR3 mediate the onset of the intracellular Ca^2+^ oscillations in the endothelial cells budding from dorsal aorta and posterior cardinal vein, respectively. Intriguingly, the endothelial Ca^2+^ waves spread along the dorsal aorta until they become restricted to the selected tip cell and, thereafter, to the stalk cells that trail behind, while Dll4/Notch signaling is responsible for suppressing the Ca^2+^ spikes in endothelial cells in close proximity to tip cells during budding [184]. VEGF-induced intracellular Ca^2+^ oscillations in tip cells are sustained by InsP_3_-dependent Ca^2+^ release and SOCE and maintain Dll4/Notch signaling to coordinate vascular morphogenesis (Table 1) [26]. These reports, therefore, confirmed for the first time that endothelial Ca^2+^ signaling supports the pro-angiogenic response to VEGF in vivo and pave the way for novel investigations employing super resolution imaging techniques to visualize endothelial Ca^2+^ signals in physiologically relevant contexts. 

### 3.4. Modulation of VEGF-Induced Endothelial Ca^2+^ Signals 

The endothelial Ca^2+^ response to VEGF undergoes complex modulation by a plethora of regulatory mechanisms. For instance, the shape of VEGF-induced endothelial Ca^2+^ signals and their impact on the angiogenic process may be modulated by S-glutathiolation of SERCA2b in HAEC [47]. A number of investigations revealed that VEGF (as well as physiological doses of exogenous hydrogen peroxide or H_2_O_2_) stimulated NADPH to generate H_2_O_2_, which in turn boosts SERCA2b activity by adding S-glutathione adducts at cysteine 674 (C674) [47]. Likewise, VEGF-induced eNOS activation may lead to SERCA2b C674 S-glutathiolation and increase SERCA2-dependent ER Ca^2+^ uptake [48]. Of note, adduction of S-glutathione at C674 favors VEGF-induced Ca^2+^ release through RyR and Orai1, thereby promoting HAEC migration and tubular network formation [47,48]. It has been further reported that, under hypoxic conditions, the increase in reactive oxygen and nitrogen species also leads to SERCA2b S-glutathiolation and endothelial tube formation [49]. In agreement with these observations, mouse hind limb ischemia induces the formation of S-glutathione adducts on endothelial SERCA2b, which is indispensable to support blood flow recovery [50]. Similar to HAEC, VEGF-induced endogenous Ca^2+^ release and extracellular Ca^2+^ entry, as well as cell migration, are impaired in mouse cardiac endothelial cells devoid of half C674, which prevents SERCA2b S-glutathiolation [50].

The angiogenic activity of VEGF may also be regulated by PMCA. PMCA isoforms are encoded by four genes (PMCA1-4) [18], of which PMCA1 and PMCA4 are the most abundant in vascular endothelial cells [185,186]. Recent investigations demonstrated that PMCA4 activity results in a low Ca^2+^ microdomain that tethers the Ca^2+^-dependent phosphatase calcineurin beneath the plasma membrane, thereby preventing its activation by VEGF [46,187]. As a consequence, VEGF fails to induce migration and tube formation, although not proliferation, in HUVEC [46].

Besides SERCA2b and PMCA4, VEGF-induced endothelial Ca^2+^ signals and pro-angiogenic activity may be finely tuned by intermediate-conductance Ca^2+^-activated K^+^ (IK_Ca_) channels [188]. The IK_Ca_ channel, encoded by the *IKCa1* gene, is up-regulated in HUVEC pre-treated with VEGF for 48 h and the pharmacological inhibition of IK_Ca_ currents prevents capillary formation in in vivo Matrigel plug assays [188]. IK_Ca_ activation leads to the hyperpolarization of endothelial V_M_, which could enhance extracellular Ca^2+^ entry during the plateau phase of the Ca^2+^ response to VEGF [189,190]. However, it remains to be elucidated whether VEGF actually recruits IK_Ca_ channels and, vice versa, whether IK_Ca_ channels sustain VEGF-induced extracellular Ca^2+^ influx during the angiogenic activity. It should, however, be pointed out that VEGF inhibits InsP_3_-dependent ER Ca^2+^ mobilization and the Ca^2+^-dependent recruitment of IK_Ca_ channels in mouse pressurized resistance arteries [191].

The Ca^2+^ response to VEGF is also sensitive to angiogenesis inhibitors, such as angiostatin and endostatin, which are necessary for vessel pruning and regression of selected vascular branches [12]. Angiostatin comprises of the first four-kringle domains of plasminogen, which is a quite abundant protein in wound microenvironment and may be cleaved into an anti-angiogenic peptide by matrix metalloproteases (MMP) [12]. Similarly, endostatin is a 20 kDa fragment of the endothelial cell basement membrane component type XVIII collagen. Endostatin is readily cleaved from the extracellular matrix by MMP to inhibit endothelial cell proliferation and migration and inducing apoptosis [12]. Early work revealed that angiostatin and endostatin prevent VEGF-induced endogenous Ca^2+^ release by inducing InsP_3_-mediated ER Ca^2+^ release and activating a Ca^2+^-entry pathway in BAEC (Table 1) [192]. It is, therefore, conceivable that their anti-angiogenic action is, at least partly, effected by depletion of the same InsP_3_-sensitive ER Ca^2+^ store targeted by VEGF [192]. Accordingly, if the ER Ca^2+^ store has been previously depleted by angiostatin and endostatin, VEGF will fail to elicit a pro-angiogenic increase in endothelial [Ca^2+^]_i_ and to activate SOCE. The ER microenvironment is not homogenous and comprises of multiple sub-sections, each performing distinct tasks [193,194]. Therefore, the ER Ca^2+^ pool targeted by angiostatin and endostatin could not be coupled to pro-angiogenic Ca^2+^-dependent decoders. Nevertheless, the fall in [Ca^2+^]_ER_ could be propagated to the ER Ca^2+^ pool engaged by VEGF through the mechanism of Ca^2+^ tunneling, thereby impairing the subsequent Ca^2+^ response to pro-angiogenic stimulation [195]. In addition, as EL Ca^2+^ mobilization has also been involved in VEGF-induced Ca^2+^ signaling, as discussed in Section 3.2, it would be interesting to assess whether angiostatin and endostatin also affect NAADP-induced endogenous Ca^2+^ mobilization. This finding was not confirmed in the intact endothelium of freshly isolated bovine coronary arteries challenged with bradykinin upon 1 h exposure to endostatin [196]. It would be interesting to assess whether endostatin and angiostatin selectively affect the VEGF-sensitive compartment of the InsP_3_-releasable ER Ca^2+^ pool. In addition, computational simulations recently investigated how the angiogenesis inhibitor, thombospondin-1, affects VEGFR2 signaling [197]. Thrombosponin-1 is a matricellular protein which binds to the extracellular matrix and inhibits endothelial cell proliferation and migration besides promoting endothelial apoptosis and repressing VEGF activity [12]. In silico investigations suggest that thrombospondin-1 binds to the endothelial receptor CD47, which in turn induces VEGR2 degradation and impairs VEGF-induced recruitment of Ca^2+^-sensitive pro-angiogenic pathways [197]. Finally, VEGF-induced endothelial Ca^2+^ signals are sensitive to Sprouty4 [198], a membrane-bound inhibitor of the ERK pathway. Sprouty4 prevents PIP_2_ hydrolysis by PLCγ1, thereby suppressing the synthesis of both InsP_3_ and DAG and dampening protein kinase C (PKC) activation, which contributes to the recruiting of ERK along with the increase in [Ca^2+^]_i_ [198].

### 3.5. Endothelial Ca^2+^ Signals Induced by bFGF, Epidermal Growth Factor (EGF), PDGF, and SDF-1α

As expected by their ability to bind to RTK coupled to the PLCγ1/InsP_3_ signaling axis, many other growth factors can induce pro-angiogenic Ca^2+^ signals in vascular endothelial cells [41,199,200]. For instance, bFGF, also known as FGF-2, triggers an increase in [Ca^2+^]_i_ in a variety of vascular endothelial cells, including HUVEC [63,199,201], SV40-transfected human corneal endothelial cells (SV40-HCEC) [202], and BAEC [192,200]. The Ca^2+^ response to bFGF is initiated by FGF receptor-1 (FGFR-1) [179,203], requires PLCγ1 activation [201], although the contribution of PLCβ3 has also been reported [63], and is essentially mediated by extracellular Ca^2+^ entry [200]. It has been suggested that bFGF-induced extracellular Ca^2+^ entry is mediated by arachidonic acid through a yet to be clearly identified Ca^2+^-permeable route [204]. The available evidence hints at a heteromeric channel formed by TRPC1 and TRPC4 in BAEC [203,205]. Of note, the pharmacological inhibition of bFGF-induced Ca^2+^ influx with carboxyamidotriazole and of arachidonic acid production by means of multiple inhibitors of phospholipase A2 and DAG lipase suppress BAEC proliferation [204]. Likewise, insulin-like growth factor-I (IGF-I) induces extracellular Ca^2+^ entry through the same pathway as that recruited by bFGF. Nevertheless, the physiological outcome of the Ca^2+^ response to IGF-I has not been evaluated [200]. Similar to VEGF, bFGF-induced Ca^2+^ entry could be modulated by the concomitant recruitment of large-conductance Ca^2+^-activated K^+^ (BK_Ca_) channels [206,207] and IK_Ca_ channels [188]. The regulatory role of endothelial V_M_ hyperpolarization is highlighted by the inhibition of bFGF-induced proliferation upon blockade of either BK_Ca_ [206] or IK_Ca_ channels [188]. Furthermore, bFGF increases the amplitude of the inwardly rectifying K^+^ current (K_IR_) in HUVEC, thereby boosting V_M_ hyperpolarization and cell proliferation [208]. Also acidic FGF (aFGF), alternately known as FGF-1, has been shown to promote extracellular Ca^2+^ entry in SV40-HCEC [202]. This investigation suggested that SV40-HCEC express L-type voltage-gated Ca^2+^ channels (VGCC) and that voltage-gated Ca^2+^ entry is induced by aFGF via FGFR-1 activation. However, pharmacological blockade of L-type VGCC also leads to SOCE inhibition, so that the exact mechanism(s) responsible for the Ca^2+^ response to aFGF remains unclear.

Early studies revealed that EGF is able to trigger a biphasic increase in [Ca^2+^]_i_ in HUVEC [199] and intracellular Ca^2+^ oscillations in rat cardiac microvascular endothelial cells (CMEC) [209], while it fails to elevate the [Ca^2+^]_i_ in ovine uterine endothelial cells [173]. Pharmacological manipulation demonstrated that EGF-induced repetitive Ca^2+^ spikes are patterned by the dynamic interplay of InsP_3_-induced ER Ca^2+^ release and SOCE, while RyR are not involved (Table 1). This investigation suggested that the oscillatory Ca^2+^ signal is required for rat CMEC proliferation as each cytosolic Ca^2+^ spike causes an increase in nuclear Ca^2+^ concentration [209]. As in other cell types, the endothelial Ca^2+^ response to EGF is likely to be mediated by Erb1, the classical EGF receptor (EGFR) [210]. PDGF, in turn, has been shown to induce multiple intracellular Ca^2+^ waveforms in PAEC: either a monophasic increase in [Ca^2+^]_i_, a biphasic Ca^2+^ signal or repetitive Ca^2+^ spikes [211]. PDGF-induced intracellular Ca^2+^ signals are initiated by PDGF β-receptor and are maintained by extracellular Ca^2+^ entry [211]. Nevertheless, the molecular machinery responsible for the Ca^2+^ response to PDGF remains to be elucidated. Finally, the chemokine SDF-1α plays a crucial role during angiogenesis (and vasculogenesis) by inducing proliferation and migration in vascular endothelial cells through the G_i_-protein coupled receptor C-X-C chemokine receptor type 4 (CXCR4) [212]. In addition, SDF-1α promotes reendothelialization of denudated arteries upon vascular injury [213]. SDF-1α has long been known to induce biphasic Ca^2+^ signals in HUVEC [214] and HRMEC [215]. Similar to PDGF, it is still unclear how SDF-1α increases the [Ca^2+^]_i_, although it has been shown that extracellular Ca^2+^ entry is involved in the onset of the signal and in SDF-1α -induced migration in HRMEC [215]. Moreover, the Ca^2+^ response to SDF-1α is finely tuned by BK_Ca_ channels, which support SDF-1α -induced proliferation and migration in HUVEC [67].

### 3.6. Endothelial Ca^2+^ Signals Induced by Angiopoietins (ANG)

While growth factors, such as VEGF, bFGF and PDGF, are required to promote sprouting angiogenesis, the subsequent maturation step is finely tuned by ANG [7]. The human ANG family comprises of three ligands, ANG-1, ANG-2 and ANG-4, and two receptors, Tie1 and Tie2. ANG-1, which is the physiological ligand of the RTK Tie2, maintains endothelial quiescence and prevents apoptosis; in addition, ANG-1 favors vessel stabilization by recruiting pericytes and smooth muscle cells and inducing the basement membrane deposition [216]. ANG-2, in turn, may function as a competitive antagonist or agonist of ANG-1 in a context-dependent manner. For instance, ANG-2 interferes with ANG-1-Tie signaling in resting endothelium to destabilize the vasculature, while it stimulates Tie2 in already activated or stressed endothelial cells [216]. Finally, Tie1 is an orphan receptor which could reduce Tie2 phosphorylation and downstream signaling [7]. A recent investigation revealed that both ANG-1 and ANG-2 induce biphasic Ca^2+^ signals in HUVEC by selectively promoting ER Ca^2+^ release through InsP_3_R and RyR (Table 1) [217]. Of note, ANG-1-induced cell migration is sustained by InsP_3_R, while ANG-2 requires both InsP_3_R and RyR. However, while ANG-1 induces in vitro tubulogenesis in a Ca^2+^-dependent manner, intracellular Ca^2+^ signaling is dispensable for ANG-2-induced tube formation [217]. Intriguingly, the lysosomal Ca^2+^ pool, which is involved in VEGF-induced Ca^2+^ signals [104], is not involved in the Ca^2+^ response to ANG-1 and ANG-2 [217]. It is, therefore, evident that distinct components of the endothelial Ca^2+^ toolkit may be recruited by the pro-angiogenic cues that regulate different steps of the angiogenic process. However, while an increase in [Ca^2+^]_i_ underlies the angiogenic effect of ANG, Tie2 activity is negatively modulated in a Ca^2+^-dependent manner. Accordingly, pre-treating HUVEC with the Ca^2+^-ionophore ionomycin promotes calmodulin (CaM) binding to the COOH-terminal tail of Tie2, which results in receptor dephosphorylation and aberrant in vivo angiogenesis [218]. Future work will have to challenge the hypothesis that the Ca^2+^-dependent regulation of Tie2 activity is accomplished by Ca^2+^-permeable routes specifically tailored to suit this function. 

## 4. The Ca^2+^-Dependent Decoders of Angiogenesis

Following the increase in [Ca^2+^]_i_, multiple Ca^2+^-dependent decoders translate endothelial Ca^2+^ signals into a pro-angiogenic outcome. These include, but are not limited to (Figure 2, Figure 3 and Figure 4): ERK 1/2 [77,104,219], PI3K/Akt [37,104,177], Ca^2+^/CaM-dependent protein kinase II (CaMKII) [220,221], calpain [222], myosin light chain kinase (MLCK) [25,223], Pyk2 [59,224], eNOS [73,104,225,226], and the transcription factors cAMP responsive element binding protein (CREB), nuclear factor of activated T-cells (NFAT) and nuclear factor kappa enhancer binding protein (NF-κB) [25,45,58,227,228,229]. VEGF has been the most frequent growth factor exploited to investigate how endothelial Ca^2+^ signaling regulates angiogenesis [24]. However, recent reports demonstrated that also non-canonical WNT signaling stimulates vessel remodeling in a Ca^2+^-dependent manner [230,231].

### 4.1. The ERK 1/2 Pathway

The ERK 1/2 phosphorylation cascade represents the most widespread signal transduction pathway whereby VEGF stimulates endothelial cell proliferation, migration and survival, promotes vascular homeostasis and regulates arterial specification [24]. Early work demonstrated that VEGF-induced increase in endothelial [Ca^2+^]_i_ stimulates the Ca^2+^-dependent PKCβ2 (cPKCβ2), which in turn recruits the downstream RAF1–MEK–ERK1/2 cascade more efficiently than Ras (Figure 2) [24,232,233]. A number of subsequent reports confirmed that VEGF requires intracellular Ca^2+^ signaling to engage the ERK1/2 pathway in vascular endothelial cells [26,77,104,171,198,199,234]. The Ca^2+^-dependent recruitment of the ERK 1/2 pathway may be sustained by endogenous Ca^2+^ release through InsP_3_R [26,177] and TPC2 [104] and by extracellular Ca^2+^ entry through TRPC3 and NCX1 [77], TRPC4 [171] and the SOCE pathway [26].

### 4.2. The PI3K/Akt Pathway 

The serine/threonine Akt1-4 kinases regulate cell proliferation, survival and resistance to apoptosis. More specifically, Akt1 is necessary to promote adult and pathological angiogenesis as well as to regulate vascular development and metabolism [24,235]. Canonically, VEGFR-2 activates PI3K via either Src and vascular endothelial cadherin [236] or the RTK Axl [237], thereby generating phosphatidylinositol-3,4,5-trisphosphate (PIP_3_), the lipid second messenger responsible for Akt activation. However, VEGF may engage the PI3K/Akt pathway also through an increase in [Ca^2+^]_i_, as demonstrated in HUVEC [104] and HCAEC [177]. Notably, TPC2 [104], InsP_3_R [170,177] and TRPC4 [171] may deliver the Ca^2+^ signal required for PI3K/Akt activation by VEGF. As discussed in [41], endothelial Ca^2+^ signaling could activate the small GTPase Ras, which in turn recruits PI3K (as well as ERK1/2) [238]. Furthermore, an increase in [Ca^2+^]_i_ could directly recruit the PI3K class II α-isoform [239]. However, the Ca^2+^-sensitive decoder coupling endothelial Ca^2+^ signaling to PI3K activations remains to be elucidated.

### 4.3. Calcineurin and NFAT

VEGF-induced endothelial Ca^2+^ signaling may also recruit the Ca^2+^-sensitive transcription factor, NFAT (Figure 2), which regulates a transcription program crucial for endothelial cell proliferation, migration, and neovessel formation [24,240]. Furthermore, NFAT is indispensable for patterning the developing vasculature [241]. Five NFAT isoforms have been described: NFAT1-NFAT5 [240]. NFAT is selectively activated by store-operated Ca^2+^ influx through Orai1 channels [242,243]. Orai1-mediated extracellular Ca^2+^ entry engages the Ca^2+^/CaM-dependent phosphatase calcineurin, which dephosphorylates multiple phosphoserines in NFAT1-NFAT4 (Figure 2), thereby promoting its nuclear translocation [24,240,242,243]. Accordingly, it has recently been demonstrated that Orai1 stimulates NFAT nuclear translocation also in vascular endothelial cells [244]. A multitude of genes involved in blood vessel formation are under transcriptional control by NFAT [58,245], including regulator of calcineurin 1 (*RCAN1*) [246] and the transcription factors, early growth response (EGR)-1 [247], EGR-3 [248] and NR4A [245], which could drive secondary gene regulatory events involved in the response to VEGF. Conversely, *RCAN1* encodes for a protein known as Down syndrome critical region 1 (DSCR1), which inhibits calcineurin activity [249], thereby hampering VEGF-induced gene expression [176], migration and angiogenesis [250,251]. In addition, DSCR1 finely tunes tubular morphogenesis by promoting VEGFR2 internalization and dampening VEGF-induced cytoskeletal reorganization and cell polarity during endothelial migration [251]. As mentioned in Section 3.3, a recent investigation demonstrated that low nanomolar doses of VEGF stimulate PAEC to undergo proliferation by selectively inducing the nuclear translocation of NFAT2 [25], the NFAT isoform responsible for cell proliferation [252]. This report failed to detect NFAT2 nuclear translocation in migrating cells when VEGF was applied at higher doses and induced a biphasic Ca^2+^ signal [25]. This finding is somehow unexpected as it has long been known that NFAT2 also controls endothelial cell migration and tube formation [253]. Solving this discrepancy will certainly deserve future studies.

### 4.4. CaMKII

CaMKII is an established decoder of intracellular Ca^2+^ signals in brain [254] and heart [255], in which it integrates repetitive oscillations in [Ca^2+^]_i_ to, respectively, induce long-term potentiation and regulate cardiac contractility. Emerging evidence has postulated a key role for endothelial CaMKII in the control of endothelial cell proliferation, migration and permeability (Figure 3) [256]. Of the four mammalian CaMKII isoforms (α,β, γ, and δ), CaMKIIα and CaMKIIβ are restricted to the brain, while CaMKIIγ and CaMKIIδ are expressed in peripheral tissues and in vascular endothelial cells [256]. Nevertheless, CaMKIIα/β heteromultimers were detected at the myo-endothelial projections of native mouse mesenteric artery endothelial cells, where they could modulate vascular tone by inhibiting InsP_3_-dependent Ca^2+^ pulsars [257]. Early work showed that CaMKII mediates VEGF-induced proliferation, migration, tubulogenesis and sprout formation in BREC [170]. Accordingly, a subsequent report confirmed that bFGF, IGF-1, and hepatocyte growth factor (HGF) induce BREC to undergo in vitro angiogenesis by recruiting CaMKII [220]. Of note, CaMKII stimulates BREC by engaging intermediate kinases to phosphorylate multiple effectors, including Akt, JNK, Src and FAK [220]. Finally, genetic deletion of CaMKIIγ and CaMKIIδ interferes with choroidal neovascularization and hypoxia-induced angiogenesis in vivo [220]. Therefore, distinct CaMKII isoforms could fulfill different functions in vascular endothelial cells: CaMKIIα and CaMKIIβ regulate the vascular tone, while CaMKIIγ and CaMKIIδ promote angiogenesis. CaMKII has also been shown to promote coronary angiogenesis by inducing capillary growth following repeated transient ischemia in mice [221]. Hypoxia indeed causes CaMKII activation, which in turn drives mouse CMEC proliferation and migration [221]. The emerging role of the endothelial CaMKII is further highlighted by a recent investigation, which showed that Notch signaling necessary for vascular remodeling and embryonic survival is compromised in a transgenic mice expressing an oxidation-resistant CaMKIIδ and lacking Regulator of G protein signaling 6 [258].

### 4.5. MLCK, Calpain and Proline-Rich Tyrosine Kinase-2 (Pyk2)

Endothelial Ca^2+^ signals drive cell migration by regulating the activity of the Ca^2+^-sensitive effectors MLCK and calpain [259,260,261]. Directional migration of stalk cells towards the source of growth factors requires the establishment of a front-to-rear polarity along their axis of movement. Migrating cells extend spike-like filopodia or broad lamellipodia, which are driven by actin polymerization, at the leading edge; these protrusions are stabilized through adhesion to the extracellular matrix or to adjoining cells via transmembrane receptors coupled to actin cytoskeleton. These adhesions serve as traction points over which cells move in the direction of the chemotactic stimulus. Thereafter, contraction of the actomyosin network causes dismantling of cell adhesions and retraction of the tail at the rear end and pulls the cell body forward [262]. As widely described elsewhere [212,261], an increase in [Ca^2+^]_i_ is required to activate MLCK, which induces myosin II-based actomyosin contraction, and the Ca^2+^-dependent protease, calpain [263] that effects the cleavage of focal adhesion proteins, including focal adhesion kinases, integrins, talin and vinculin. In addition, intracellular Ca^2+^ signals recruit the Ca^2+^-sensitive tyrosine kinase Pyk2, which sustains cytoskeletal reorganization at nascent adhesion sites and VEGF-induced eNOS activation [59,224]. Surprisingly, VEGF-induced intracellular Ca^2+^ release is compromised in Pyk2-deficient mouse aortic endothelial cells [59], thereby suggesting that Pyk2-dependent phosphorylation tunes the endothelial Ca^2+^ toolkit.

Similar to other cell types, migrating endothelial cells present a remarkable Ca^2+^ gradient along their axis of movement, as cytosolic Ca^2+^ is higher at the rear edge due to the higher PMCA activity at the leading edge [223,264]. However, local Ca^2+^ pulses selectively arise at the front rear of migrating endothelial cells uniformly exposed to bFGF, as recently shown in HUVEC [260]; bFGF-induced local Ca^2+^ flickers are driven by local PLCγ1 activation and InsP_3_-dependent Ca^2+^ release followed by SOCE activation (Table 1) and are necessary to engage MLCK (and, probably, calpain) in the front of migrating cells [223,263]. Notably, SOCE activation at the leading edge is boosted by the local decrease in [Ca^2+^]_ER_, which favors STIM1 redistribution at ER-plasma membrane junction at the front of migrating cells [223]. These findings are somehow different as compared to those reported in PAEC, in which MLCK is recruited by a biphasic Ca^2+^ signal [25]. This discrepancy might reflect differences in the vascular bed (umbilical vein vs. aorta) or species (human vs. porcine). Nevertheless, repeated Ca^2+^ flickers spatially restricted at the rear part of the cell are regarded as a hallmark of migrating cells and could also be sustained by extracellular Ca^2+^ entry through the stretch-sensitive TRP Melastatin 7 (TRPM7) [259]. Of note, shear stress may induce Ca^2+^ entry and downstream recruitment of calpain at focal adhesions of migrating HUVEC [263,265], although TRPM7 silencing does not affect migration [266] and calpain activation [267].

**Table 1 ijms-20-03962-t001:** The endothelial Ca^2+^ toolkit: Recruitment by pro- and anti-angiogenic cues.

Signal	EC	InsP_3_R	RyR	TPC	STIM1/Ora1	TRPC1/TRPC4	TRPC3/TRPC6	Ref.
VEGF	HUVEC	Yes	N.I.	Yes	Yes	Yes	Yes	[73,76,77,104,122,123,124,163,164]
VEGF	EA.hy926	Yes	N.I.	N.I.	No	No	No	[165]
VEGF	HAEC	Yes	Yes	N.I.	Yes	N.I.	N.I.	[48]
VEGF	Zebrafish tip cells	Yes	N.I.	Yes	N.I.	N.I.	N.I.	[26]
bFGF	HUVEC	Yes	N.I.	N.I.	Yes	N.I.	N.I	[203,205,223,260]
EGF	CMEC	Yes	No	N.I.	Yes	N.I.	N.I.	[209]
ANG	HUVEC	Yes	Yes	No	No	No	No	[217]
Angiostatin	BAEC	Yes	N.I.	N.I.	N.I.	N.I.	N.I.	[192]
Endostatin	BAEC	Yes	N.I.	N.I.	N.I.	N.I.	N.I.	[192]

Abbreviations: BAEC = bovine aortic endothelial cells, CMEC = cardiac microvascular endothelial cells; HAEC = human aortic endothelial cells; HUVEC = human umbilical vein endothelial cells. Only investigations assessing more than one pro-angiogenic Ca^2+^ entry/release pathway were described.

### 4.6. eNOS

It has long been known that NO plays a crucial role in VEGF-induced angiogenesis by stimulating endothelial cell proliferation, migration, substrate adhesion, hyperpermeability and tube formation [59,234,268,269]. Endothelial eNOS is mainly sequestered within Ω-shaped invaginations of the plasma membrane, known as caveolae, which are enriched in cholesterol and are coated on their cytosolic surface by caveolin-1 (Cav1) [270]. In quiescent cells, eNOS activation is hindered by the physical interaction with Cav1, which prevents CaM association to the CaM-binding sequence located between the NH_2_-terminal oxygenase and the inhibitory COOH-terminal reductase domain. However, an increase in [Ca^2+^]_i_ displaces Cav1 from eNOS, thereby relieving this tonic inhibition and inducing NO release (Figure 4) [270]. 

Alternately, eNOS activity can be stimulated by phosphorylation of at least three residues (S615, S633, and S1177) located within the autoinhibitory reductase domain, which prevents eNOS activation in the absence of a Ca^2+^ rise [270]. A number of kinases stimulate eNOS, including Akt, protein kinase A (PKA), PKC, CaMKII, AMP-activated kinase (AMPK), and Pyk2 [59,270,271]. Early work demonstrated that VEGF induces immediate eNOS activation through an increase in [Ca^2+^]_i_ followed by delayed NO synthesis which requires eNOS phosphorylation by Akt and PKC [225,226]. The heat shock protein 90 is required to recruit Akt to eNOS upon VEGF stimulation and sustain NO release after the increase in [Ca^2+^]_i_ [225]. In addition, VEGF may phosphorylate eNOS and promote NO-dependent angiogenesis by engaging AMPK. Of note, VEGF-induced AMPK activation is sustained by the Ca^2+^/CaM-dependent kinase kinase [271]. Whereas the role of Ca^2+^ in the immediate eNOS activation has been long established, InsP_3_-dependent ER Ca^2+^ release has been regarded as the main pathway responsible for VEGF-induced endothelial NO release [225,226,272]. However, VEGF-induced eNOS may also be recruited by Ca^2+^ release through TPC2 [104] and by SOCE [272], which represents the most widespread pathway to induce endothelial NO release [120]. 

### 4.7. Non-Canonical Wnt/Ca^2+^ Signaling Pathway

Non-canonical Wnt/Ca^2+^ signaling pathway plays a pivotal role during early development by controlling cell proliferation, migration, polarity and cell fate specification. In addition, dysregulation of the Wnt/Ca^2+^ signaling pathway may also lead to neoplastic transformation [273]. Non-canonical Wnt ligands, such as Wnt5a, bind to specific cell surface Frizzled (Fz) receptors, i.e., Fz2-Fz6, which are G_q/11_PCR able to recruit PLCβ and induce InsP_3_-dependent Ca^2+^ release [273]. The following increase in [Ca^2+^]_i_ stimulates calcineurin and CaMKII to, respectively, engage NFAT and NF-κB, thereby inducing gene expression [273]. Recent investigations provided the evidence that Wnt5a controls vascular morphogenesis by fine-tuning the angiogenic process [230,231,274,275,276]. Wnt5a may be secreted by vascular endothelial cells [276] or by angiogenic myeloid cells, such as monocytes [277] and macrophages [278]. For instance, autocrine Wnt5a stimulates HUVEC to undergo angiogenesis by activating CaMKII [279]. Likewise, monocyte-derived Wnt5a recruits NF-κB in HDMEC to induce migration, tube formation and neovessel formation in vivo [277]. Furthermore, autocrine Wnt5a/Ca^2+^ signaling induces NFAT-dependent gene expression to protect HUVEC from apoptosis, prevent vascular regression in retina and drive vascular morphogenesis in aorta [231]. Finally, the Wnt signaling mediator, Secreted frizzle-related protein 2 (SFRP2), activates NFAT to promote migration and tube formation in HCAEC and boosts angiogenesis in a mouse model of angiosarcoma [230]. Of note, recombinant Wnt5s induces intracellular Ca^2+^ signals in HDMEC, although the involvement of InsP_3_-dependent ER Ca^2+^ release remains to be elucidated.

## 5. The Role of Ca^2+^ Signaling in Vasculogenesis

While the role of endothelial Ca^2+^ signaling in angiogenesis has long been recognized, recent reports provided the evidence that an increase in [Ca^2+^]_i_ plays a crucial role also in vasculogenesis [19,41]. It should, however, be pointed out that the term EPC actually encompasses a large repertoire of different cell types that present distinct phenotypes and stimulate angiogenesis through diverse mechanisms [9,10,280]. A recent Consensus Statement on Nomenclature proposed to abandon the term EPC in favor of a terminology that is more appropriate to define the two broad categories of cell types involved in neovessel formation [280]. These include hematopoietic and endothelial progenitors, which are known, respectively, as myeloid angiogenic cells (MAC) and endothelial colony forming cells (ECFC). MAC, also termed circulating angiogenic cells (CAC) or “early” EPC, are actually hematopoietic progenitors which are liberated from the bone marrow and stimulate capillary sprouting in a paracrine manner. Conversely, ECFC, also known as blood outgrowth endothelial cells (BOEC) or late outgrowth EPC, truly belong to the endothelial lineage, reside in vascular stem cell niches, show high clonal potential, form bidimensional capillary networks in vitro and integrate into host vasculature in vivo [280,281,282]. Accordingly, ECFC may be instrumental to vascular reconstruction after an ischemic insult [280,281,282] or mediate the angiogenic switch in growing tumors [283,284]. Therefore, herein we mainly focus on the role of intracellular Ca^2+^ signaling in driving ECFC’s angiogenic activity. Most of the findings described in the following chapters were carried out in circulating ECFC, i.e., ECFC isolated from peripheral blood, unless otherwise stated.

### 5.1. The Ca^2+^ Toolkit in Human ECFC: Endogenous Ca^2+^ Release and Extracellular Ca^2+^ Entry

The network of Ca^2+^-clearing systems that contribute to maintain the resting [Ca^2+^]_i_ in ECFC and to remove cytosolic Ca^2+^ upon chemical stimulation is similar to that described in vascular endothelial cells (Figure 5) [19,20,41]. Briefly, an in-depth transcriptomic analysis revealed that circulating ECFC express SERCA2b, PMCA1b and PMCA4b, while they are devoid of the endothelial NCX isoforms, such as NCX1.3 and NCX1.7 [285]. The Ca^2+^ response to pro-angiogenic growth factors and chemokines is shaped by both endogenous Ca^2+^ release and extracellular Ca^2+^ entry (Figure 5). Intracellular Ca^2+^ release is sustained by ER Ca^2+^ release through InsP_3_R1-3 [286], but not RyR [287,288], and EL Ca^2+^ mobilization through TPC1 [287]. InsP_3_ is synthesized following PLCβ2 activation [289], while it is unclear which PLCγ isoform is expressed in ECFC. The main pathway responsible for extracellular Ca^2+^ entry is provided by SOCE [135,288], which is mediated by the interplay between STIM1, Orai1 and TRPC1 [288,290]. Quite surprisingly, circulating ECFC lack DAG-gated TRPC3 and TRPC6 [288], although the former is abundantly expressed in umbilical cord blood (UCB)-derived ECFC [291]. Pro-angiogenic Ca^2+^ signals may be delivered by TRP Vanilloid 4 (TRPV4) [292], which is gated by arachidonic acid [287]. It is, however, still unknown whether TRPV4 is also engaged by growth factors or chemokines.

### 5.2. IGF-2 and SDF-1α Stimulate ECFC Homing to Hypoxic Tissues through an Increase in [Ca^2+^]_i_


Ischemic tissues liberate multiple chemokines, such as IGF-2 and SDF-1α, which recruit circulating ECFC or promote ECFC mobilization in peripheral circulation and stimulate their physical engraftment within nascent neovessels [289,293]. Early work showed that IGF-2 binds to IGF receptor 2 (IGFR2) to activate PLCβ2 and induce InsP_3_-depedendent ER Ca^2+^ release in UCB-derived ECFC [289]. The ensuing increase in [Ca^2+^]_i_ is necessary to instigate UCB-derived ECFC to migrate and to adhere to a fibronectin matrix in vitro. Furthermore, IGF-2-evoked Ca^2+^ signaling drives UCB-derived ECFC homing to damaged vessels and engraftment into nascent vasculature in vivo, thereby promoting vascular reconstruction in a murine model of hindlimb ischemia [289]. Likewise, SDF-1α elicits a biphasic increase in [Ca^2+^]_i_ in circulating ECFC [294,295]. The Ca^2+^ response to SDF-1α is triggered by CXCR4, initiated by InsP_3_-dependent ER Ca^2+^ mobilization and sustained by SOCE. The interaction between InsP_3_Rs and SOCE recruits the PI3K/Akt and ERK1/2 signaling pathways, which in turn regulate ECFC migration in vitro and neovessel formation in vivo [294].

### 5.3. VEGF and NAADP Stimulate Proliferation and Tube Formation through an Increase in [Ca^2+^]_i_ in ECFC

VEGF is massively secreted by ischemic tissues to stimulate local endothelial cells to undergo angiogenesis and circulating ECFC to integrate within the damaged vascular network [296,297]. It has been shown that VEGF binds to VEGFR2 to induce repetitive oscillations in [Ca^2+^]_i_ in circulating [286] and UCB-derived [291] ECFC. VEGF-induced intracellular Ca^2+^ oscillations in circulating ECFC are shaped by rhythmic ER Ca^2+^ release through InsP_3_Rs followed by SERCA-mediated Ca^2+^ sequestration into ER lumen. SOCE is required to sustain the spiking response by refilling the ER with Ca^2+^ during prolonged stimulation [286]. Notably, VEGF-induced intracellular Ca^2+^ oscillations are more robust and frequent in UCB-derived ECFC [282,291] and this feature could result in their higher sensitivity to VEGF [298]. Unlike their circulating counterparts, UCB-derived ECFC express TRPC3 and TRPC3-mediated extracellular Ca^2+^ entry triggers the dynamic interplay between InsP_3_ and SOCE (Figure 5) [291]. It has, therefore, been proposed that TRPC3 over-expression could represent an alternative strategy to rejuvenate the reparative phenotype of aging ECFC and improve the outcome of therapeutic angiogenesis in cardiovascular patients [41,139]. VEGF-induced intracellular Ca^2+^ oscillations stimulate proliferation and in vitro tubulogenesis by promoting the nuclear translocation of NF-κB [286]. The intermediate Ca^2+^-sensitive decoder that translates the spiking Ca^2+^ signal into NF-κB activation is likely to be provided by CaMKII, which controls ECFC proliferation [299] and is sensitive to endothelial Ca^2+^ oscillations [300]. Notably, VEGF-induced SOCE is dramatically enhanced at day 0, but not at days 7 and 180, from the event in circulating ECFC harvested from peripheral blood of subjects who underwent acute myocardial infarction. Furthermore, ECFC frequency is higher within the first 24 h after an acute myocardial infarction as compared to healthy individuals, although their angiogenic activity is not altered [301]. Therefore, it is tempting to speculate that SOCE up-regulation results in the higher ECFC frequency reported in infarcted patients [301]. 

In addition to VEGF, exogenous delivery of NAADP through a liposomal formulation has been shown to stimulate ECFC proliferation [302]. NAADP acts by releasing EL Ca^2+^ through TPC1 (Figure 5) [302,303] and preliminary evidence suggested that lysosomal Ca^2+^ release could, in turn, be amplified by InsP_3_R [303]. While this observation is yet to be fully confirmed, the alternative hypothesis that InsP_3_-dependent Ca^2+^ release is necessary to refill the lysosomal Ca^2+^ pool cannot be ruled out [113,116]. In addition, NAADP could interact with InsP_3_ to initiate the rhythmic Ca^2+^ response to VEGF also in circulating ECFC.

### 5.4. VEGF-Induced Intracellular Ca^2+^ Oscillations Are Down-Regulated in Tumor-Derived ECFC

It has long been known that the Ca^2+^ toolkit is dramatically remodeled in tumor endothelial cells, thereby exacerbating the angiogenic switch and conferring resistance to anti-cancer treatments, as widely discussed elsewhere [32,304,305,306,307]. Likewise, InsP_3_-dependent ER Ca^2+^ release and SOCE are altered in tumor-derived ECFC, although their dysregulation depends on the tumor type [19,20]. For instance, the intracellular Ca^2+^ response to InsP_3_ production is impaired in renal cellular carcinoma (RCC)-derived ECFC due to the reduction in [Ca^2+^]_ER_ and InsP_3_R transcripts [285,290]. As a consequence, VEGF fails to elicit intracellular Ca^2+^ signals in RCC-derived ECFC, although VEGFR2 is normally expressed and SOCE is up-regulated due to the over-expression of STIM1, Orai1 and TRPC1 [290,308]. VEGF-induced oscillations in [Ca^2+^]_i_ are down-regulated and do not stimulate either proliferation or tube formation also in breast cancer (BC)-derived ECFC [309]. As reported in RCC-derived ECFC [290], the ER is less prone to release Ca^2+^ [309,310], although there is no change in InsP_3_R expression and SOCE is not impaired in BC-derived ECFC [309]. Furthermore, VEGF does not trigger sizeable Ca^2+^ signals in infantile hemangioma (IH)-derived ECFC [311], whereas the spiking Ca^2+^ response arises but does not stimulate any angiogenic activity in primary myelofibrosis (PMF)-derived ECFC [308]. It has, therefore, been suggested that VEGF is actually unable to promote an angiogenic behavior in tumor-derived ECFC due to the derangement of their Ca^2+^ signaling machinery [20,283,310]. This feature could explain the resistance (primary or secondary) to anti-VEGF drugs that has long been reported in RCC, BC and PMF patients and highlights the urgency of identifying novel targets to interfere with angiogenesis in cancer patients [283,312]. Of note, pharmacological and genetic deletion of SOCE inhibits proliferation and tube formation in RCC- [290], BCC- [309], and IH-derived [311] ECFC. The growth factor(s) responsible for SOCE activation in tumor-derived is (are) still unknown, but it is worth noting that SOCE is constitutively active in IH-derived ECFC due to the partial depletion of the ER Ca^2+^ content [311].

### 5.5. The Ca^2+^ Toolkit in Rodent MAC

The pro-angiogenic role of intracellular Ca^2+^ signaling has also been evaluated in rat and mouse bone marrow-derived MAC, which stimulate angiogenesis by delivering paracrine signals, but do not physically integrate within neovessels. Early investigations demonstrated that SOCE is abated by the genetic deletion of Stim1 and TRPC1 in rat MAC, whereas the involvement of Orai1 was not investigated [313,314,315]. The genetic and pharmacological ablation of SOCE machinery suppresses proliferation, migration, and in vitro tubulogenesis in rat MAC [313,314,315]. Likewise, SOCE is expressed and drives proliferation, migration and tube formation also in murine MAC, which express STIM1, Orai1 and TRPC1 [316,317]. Although the molecular composition of SOCE was not dissected in this report, SOCE supports VEGF-induced repetitive Ca^2+^ spikes [316], as observed in human ECFC [286]. Moreover, STIM1, Orai1 and TRPC1 are down-regulated in MAC deriving from atherosclerotic mice; as a consequence, SOCE, VEGF-induced intracellular Ca^2+^ oscillations, proliferation and migration are attenuated [316]. The Ca^2+^-dependent decoder of VEGF-induced intracellular Ca^2+^ oscillations in MAC is represented by eNOS [316,317], although it is likely that future investigations reveal additional Ca^2+^-sensitive effectors.

## 6. Conclusions

Endothelial Ca^2+^ signaling is instrumental in translating pro-angiogenic cues into an effective stimulus to induce proliferation, migration, tube formation and neovessel formation. Nevertheless, the components of the Ca^2+^ handling machinery underlying the Ca^2+^ response to growth factors and chemokines may be differently assorted depending on stimulus, vascular bed and species. Shedding light on this crucial aspect of endothelial Ca^2+^ signaling requires more investigations on growth factors other than VEGF, which has largely been exploited to understand how an increase in [Ca^2+^]_i_ regulates angiogenesis and vasculogenesis. The available evidence hints at InsP_3_Rs and SOCE as the major driver of pro-angiogenic Ca^2+^ signals in both vascular endothelial cells and ECFC/MAC. Nevertheless, the role of NAADP and TPC1-2 is rapidly emerging and it remains to understand whether extracellular Ca^2+^ entry through DAG-gated channels, i.e., TRPC3 and TRPC6, during the plateau phase observed in some cell types, such as HUVEC, is merely redundant or is required to recruit Ca^2+^-sensitive decoders that are not targeted by SOCE. An additional feature that should be taken into account is the dose-response relationship of each growth factor as the Ca^2+^ signature and Ca^2+^-dependent pro-angiogenic activity could vary depending on the dose, as reported for VEGF in PAEC. 

Understanding how endothelial Ca^2+^ signaling controls angiogenesis and vasculogenesis is indispensable to design alternative strategies to induce therapeutic angiogenesis in ischemic disorders and interfere with the vascular network in solid malignancies. For instance, it has been demonstrated that genetic or pharmacological manipulation of the endothelial Ca^2+^ toolkit improves angiogenesis in vivo. VEGF-induced extracellular Ca^2+^ influx was enhanced by engineering HDMEC with an adenovirus encoding for TRPC6, thereby enhancing their proliferation rate [74]. Similarly, hypoxic preconditioning increased TRPC4 levels and boosted migration in human pulmonary artery endothelial cells [318]. These pieces of evidence suggest that it is therapeutically feasible to exploit endothelial Ca^2+^ signalling to stimulate therapeutic angiogenesis. Of note, it has been suggested that autologous or endogenous ECFCs, which represent the most suitable EPC subtype for regenerative therapy of ischemic disorders, could be redirected towards a more reparative phenotype by intervening on their Ca^2+^ toolkit. It has been suggested that ECFCs’ angiogenic activity could be improved by lentivirus-mediated expression of TRPC3, subsequent expansion in vitro and reinoculation into the ischemic tissue [139,319]. Moreover, a recent investigation demonstrated that intramyocardial injection of the secretome collected from hypoxic UCB-derived mesenchymal stem cells induced angiogenesis by promoting intracellular Ca^2+^ oscillations in resident ECFC in a murine model of AMI [320]. This evidence provided the proof of concept that it is therapeutically feasible to induce cardiac revascularization and partially rescue cardiac function by exploiting intracellular Ca^2+^ signalling in resident ECFC. An alternative, but not mutually exclusive mechanism, consists in taking advantage from small molecule drug to stimulate angiogenesis by enhancing endothelial Ca^2+^ signaling. For instance, inhibiting PMCA4 with the small molecule aurintricarboxylic acid enhanced VEGF-induced calcineurin activation, thereby boosting HUVEC migration and tube formation as well as neovessel formation in a murine model of hindlimb ischemia [45]. Conversely, the pharmacological blockade of Orai1 with carboxyamidotriazole, which targets both vascular endothelial cells [73] and tumor-derived ECFC [290], has been probed in phase I-III clinical trials launched towards several malignancies [304,312]. Although more specific Orai1 blockers are certainly required to avoid off-target effects, targeting endothelial SOCE could represent an alternative strategy to fight cancer. Likewise, the pharmacological blockade of TPC1 with NED-19 is coming of age as an efficient means to retard tumor vascularization [104,321]. Alternately, it has been suggested that endothelial Ca^2+^ signaling could be exploited to favor normalization of tumor vasculature [304], thereby improving the therapeutic outcome of classical anticancer treatments, such as radiotherapy and chemotherapy. For instance, stimulation of purinergic P2X7 and P2Y11 receptors inhibits migration in breast tumor-derived endothelial cells and induces vessel normalization in a Ca^2+^-dependent manner [33].

## Figures and Tables

**Figure 1 ijms-20-03962-f001:**
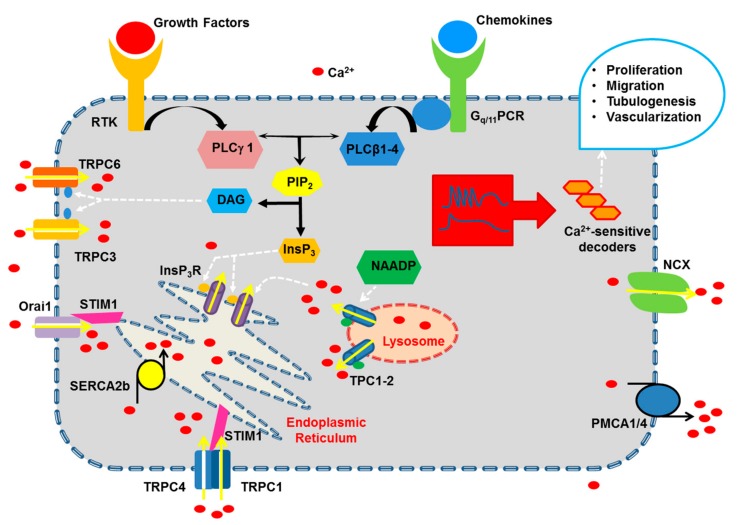
The pro-angiogenic Ca^2+^ toolkit in vascular endothelial cells. Pro-angiogenic cues, such as growth factors and chemokines, bind to specific receptor tyrosine kinases (RTK) and G_q/11_-protein Coupled Receptors (G_q/11_PCR) thereby activating multiple phospholipase C (PLC) isoforms, which in turn cleave phosphatidylinositol 4,5-bisphosphate (PIP_2_) into inositol-1,4,5-trisphosphate (InsP_3_) and diacylglycerol (DAG). InsP_3_ triggers Ca^2+^ release from the endoplasmic reticulum (ER) through InsP_3_ receptors (InsP_3_R), while DAG stimulates extracellular Ca^2+^ entry through TRPC3 and TRPC6. However, the major Ca^2+^-entry pathway in vascular endothelial cells is provided by store-operated Ca^2+^ entry (SOCE), which is mainly mediated by the physical interaction between STIM1 and Orai1. In addition, SOCE may be sustained by the interplay among STIM1, Transient Receptor Potential (TRP) Canonical 1 (TRPC1) and TRPC4, with [35] or without the involvement of Orai1. Endogenous Ca^2+^ release may also be sustained by ryanodine receptors (RyR, not shown) and by endolysosomal two-pore channel 1-2 (TPC1-2), which are gated by nicotinic acid adenine dinucleotide phosphate (NAADP). Multiple Ca^2+^-transporting systems maintain resting Ca^2+^ concentration and clear cytosolic Ca^2+^ after the pro-angiogenic signal. These include Sarco-Endoplasmic Reticulum Ca^2+^-ATPase 2a (SERCA2a), Plasma Membrane Ca^2+^-ATPase 1 (PMCA1) and PMCA4, and Na^+^/Ca^2+^ exchanger (NCX). Please, see the text for a more detailed description of how the endothelial Ca^2+^ toolkit is recruited by pro-angiogenic cues.

**Figure 2 ijms-20-03962-f002:**
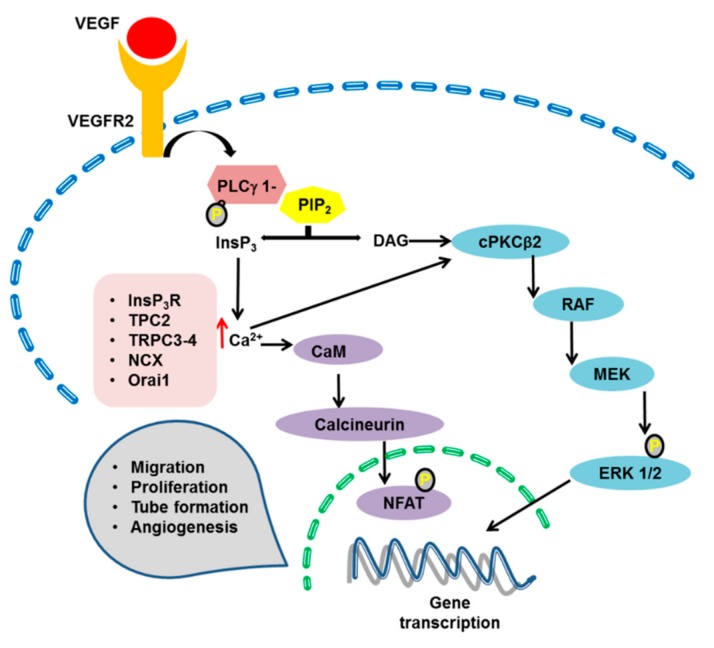
Ca^2+^-dependent activation of the extracellular signal-regulated kinase (ERK) pathway and of the Nuclear Factor of Activated T-cells (NFAT). VEGF binding to VEGFR2 triggers an increase in intracellular Ca^2+^ concentration that may stimulate the ERK1/2 phosphorylation cascade or NFAT nuclear translocation. VEGF-induced endothelial Ca^2+^ signals may be mediated by multiple Ca^2+^-entry/release pathways, depending on species and vascular bed. The Ca^2+^ response to VEGF may recruit the Ca^2+^-dependent PKCβ2 (cPKCβ2), which engages the downstream RAF1–MEK–ERK1/2 cascade to induce gene expression. An increase in [Ca^2+^]_i_ is required to promote cPKCβ2 translocation to the plasma membrane, where it is activated by DAG. Moreover, VEGF-induced endothelial Ca^2+^ signals may be sensed by calmodulin (CaM), which in turn activates calcineurin to dephosphorylate NFAT, thereby inducing its nuclear translocation. See Figure 1, Section 4.1 (ERK) and Section 4.3 (calcineurin and NFAT) for further details.

**Figure 3 ijms-20-03962-f003:**
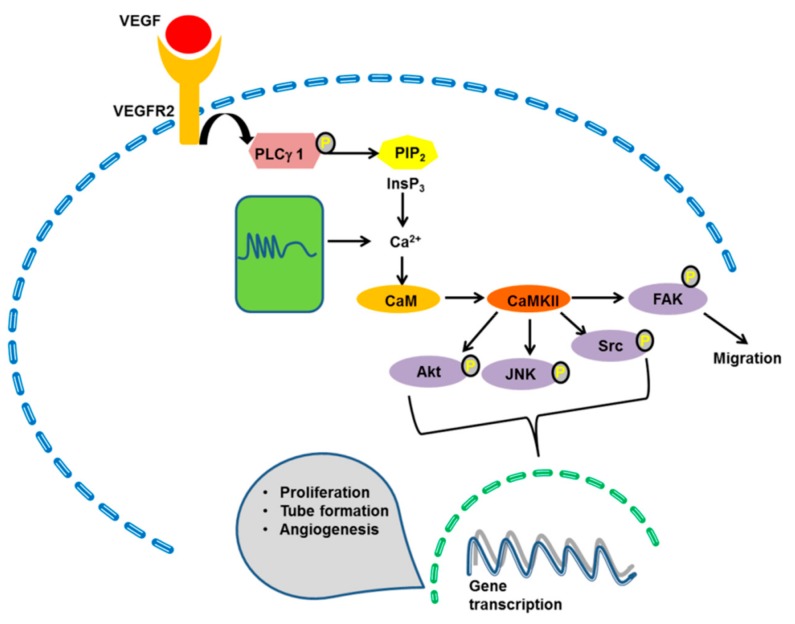
The Ca^2+^-dependent activation of Ca^2+^/Calmodulin (CaM)-dependent protein kinase 2 (CaMKII). Endothelial Ca^2+^ oscillations recruit CaMKII, which, in turns, stimulate angiogenesis by phosphorylating multiple targets, as widely illustrated in Section 4.4. VEGF-induced endothelial Ca^2+^ oscillations are sensed by CaM, which in turn stimulates CaMKII to phosphorylate multiple targets, including FAK to promote endothelial cell migration and Akt, JNK and Src to induce gene expression. The Ca^2+^ entry/release pathways that are recruited by VEGF to engage endothelial CaMKII are yet to be fully elucidated.

**Figure 4 ijms-20-03962-f004:**
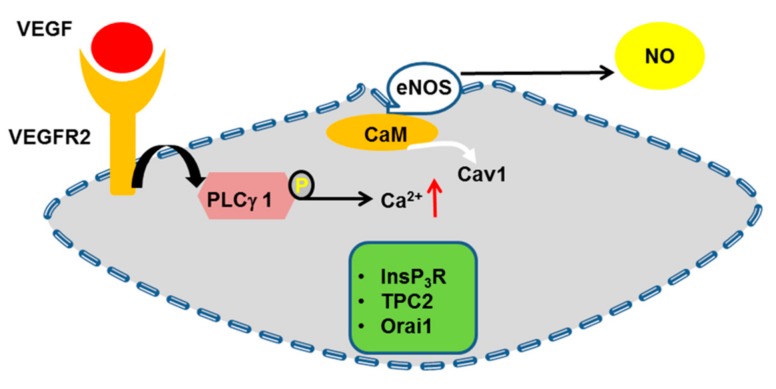
The Ca^2+^-dependent activation of the endothelial nitric oxide (NO) synthase (eNOS). VEGF binding to VEGFR2 causes an increase in intracellular Ca^2+^ concentration that displaces caveolin 1 (CaV1) from eNOS, thereby removing the tonic inhibition and inducing NO release. VEGF may impinge on several Ca^2+^ entry/release pathways to engage eNOS, including InsP_3_R, TPC2 and Orai1. See Figure 1 and Section 4.6. for further details.

**Figure 5 ijms-20-03962-f005:**
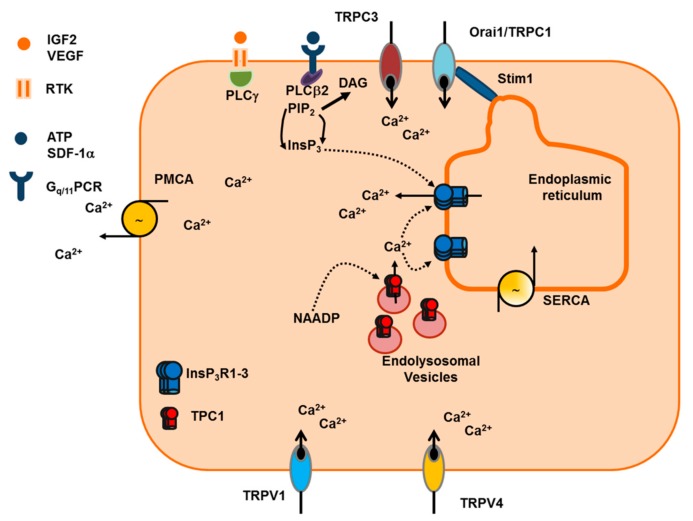
The Ca^2+^ toolkit in endothelial colony forming cells (ECFC). Growth factors, such as VEGF, and chemokines, such as stromal derived factor-1α (SDF-1α), bind to specific RTK and G_q/11_PCR, thereby activating multiple PLC isoforms, which in turn cleave PIP_2_ into InsP_3_ and DAG. InsP_3_ triggers ER-dependent Ca^2+^ release through InsP_3_R, while DAG gates TRPC3 exclusively in umbilical cord blood-derived ECFCs. SOCE is the major Ca^2+^ entry pathway also in ECFC, in which it is mediated by the dynamic interplay among STIM1, Orai1 and TRPC1. Endogenous Ca^2+^ release is further supported by NAADP, which evokes EL Ca^2+^ release through TPC1. SERCA and PMCA contribute to maintain resting Ca^2+^ levels and clear cytosolic Ca^2+^ after a pro-angiogenic Ca^2+^ signal. Pro-angiogenic Ca^2+^ signals may also be delivered by TRP Vanilloid 1 (TRPV1) and TRPV4. See Section 5.2. (SDF-1α) and Section 5.3. (VEGF and NAADP) for further details.

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
