# Peer review of "Endothelial Ca2+ Signaling, Angiogenesis and Vasculogenesis: Just What It Takes to Make a Blood Vessel"

_ijms, 2019, doi:10.3390/ijms20163962_

Round 1

Reviewer 1 Report

The review entitled Endothelial Ca2+ signaling, angiogenesis and vasculogenesis: Just what it takes to make a blood vessel by Moccai et al. provides an excellent overview of the role of Calcium in blood vessel growth. The review is well written and provides multiple examples, and the topic is important and timely.

Specific comments:

1.    Please check that each reference is associated with the relevant text
2.    Figure legends need to be modified and expanded when stating “as more widely illustrated in-text” please refer the reader to the relevant section. For example, Figure 4 includes TPC2 and Orai1, but from the relevant section of the text, it is unclear how these fit into the figure.
3.    A brief introduction to the main types of calcium signalling and their functions in different settings will benefit the reader.
4.    There are missing references in the reference list. Please check.
5.    Please check the acronyms and provide full forms when it appears for the first time in the text.
6.    Figure legends will benefit by expansion to provide full details to understand the schematics.

Author Response

Dear Referee #1,

We were happy to read your comments about our manuscript entitled: “Endothelial Ca2+ Signaling, angiogenesis and vasculogenesis: Just what it takes to make a blood vessel” for publication as Review Article in International Journal of Molecular Sciences – Special Issue Calcium Signaling in Human Health and Diseases 2.0.

We truly believe that your comments did improve the quality of this manuscript and we are grateful for the attention you paid to our work. We amended the manuscript according to your indications and addressed all the criticisms you raised. More specifically:

Please check that each reference is associated with the relevant text

We thank the Reviewer for this observation and confirm that each reference is associated with the relevant text.

Figure legends need to be modified and expanded when stating “as more widely illustrated in-text” please refer the reader to the relevant section. For example, Figure 4 includes TPC2 and Orai1, but from the relevant section of the text, it is unclear how these fit into the figure.

We thank the Reviewer for this observation. For each Figure, we specified the corresponding Section of the text. For instance, in the case of Figure 4 and TPC2, we described in Section 4.6 that VEGF can recruit eNOS by inducing TPC2-dependent intracellular Ca2+ mobilization or SOCE activation. We edited the text to make this assertion clear.

A brief introduction to the main types of calcium signalling and their functions in different settings will benefit the reader.

We thank the Reviewer for this observation. We introduced a few lines for each Ca2+-permeable channel to describe its main functions in other cellular contexts.

There are missing references in the reference list. Please check.

We thank the Reviewer for this observation. All the references were checked.

Please check the acronyms and provide full forms when it appears for the first time in the text.

We thank the Reviewer for this observation. All the acronyms were expanded when they appear for the first time. According to the journal requirements, however, each acronym was again expanded when appearing for the first time also in figure legends.

Figure legends will benefit by expansion to provide full details to understand the schematics.

We thank the Reviewer for this observation. Each figure legend has been expanded to provide more information about the activation of the Ca2+-dependent decoders.

We are grateful for your insightful revision and hope that you will not regard the manuscript worth being published on this Special Issue of the International Journal of Molecular Sciences.

With my kindest regards,

Francesco Moccia

Francesco Moccia, PhD

Laboratory of General Physiology,

Department of Biology and Biotechnology “L. Spallanzani”

University of Pavia,

Via Forlanini 6, 27100, Pavia, Italy.

Tel: 0039 0382 987614.

Fax: 0039 0382 987527.

Reviewer 2 Report

The manuscript entitled “Endothelial Ca2+ signaling, angiogenesis and vasculogenesis: Just what it takes to make a blood vessel” reviews the underlying mechanisms of pro-angiogenic Ca2+ signaling in vascular endothelial cells. The authors overview the role of endothelial Ca2+ signaling in angiogenesis and vasculogenesis induced by growth factors, chemokines and angiogenic modulators. This review provides the general understanding of a key signaling for angiogenesis regulation. However, the following issues must be addressed before the manuscript can be suitable for publication.

Comments:

To clarify the role of endothelial Ca2+ signaling in angiogenesis, the authors must additionally explain the relationship between anti-angiogenic modulators and Ca2+ signaling. Based on their overview on endothelial Ca2+ signaling, the authors must present their opinion and perspective into future therapeutic strategy for angiogenesis-related diseases in Conclusion part of this paper. In addition to each Figure, the authors need to provide a Table that generally overview the pro-angiogenic and anti-angiogenic pathways which are associated with the regulation of endothelial Ca2+ signaling. In Figure 1, the red circle should be labeled Ca2+. In 2.1.2.1 ~ 2.1.2.3, it would be better to describe in more detail the mechanisms by which Ca2+ is released by InsP3R, RyR and TPC.

Author Response

Dear Referee #2,

We were happy to read your comments about our manuscript entitled: “Endothelial Ca2+ Signaling, angiogenesis and vasculogenesis: Just what it takes to make a blood vessel” for publication as Review Article in International Journal of Molecular Sciences – Special Issue Calcium Signaling in Human Health and Diseases 2.0.

We truly believe that your comments did improve the quality of this manuscript and we are grateful for the attention you paid to our work. We amended the manuscript according to your indications and addressed all the criticisms you raised. More specifically:

1. To clarify the role of endothelial Ca2+ signaling in angiogenesis, the authors must additionally explain the relationship between anti-angiogenic modulators and Ca2+ signaling.

We thank the Reviewer for this observation. Previously, we had briefly described the mechanism whereby anti-angiogenic modulators, such as angiostatin and endostatin, may interfere with endothelial Ca2+ signaling. In the revised manuscript, we expanded upon this subject by advancing some speculations regarding the mechanism of ER Ca2+ tunneling and the inhibition of VEGF and bFGF-induced Ca2+ signals.

2. Based on their overview on endothelial Ca2+ signaling, the authors must present their opinion and perspective into future therapeutic strategy for angiogenesis-related diseases in Conclusion part of this paper.

We thank the Reviewer for this observation. We have further expanded our opinion and perspective into the therapeutic application of endothelial Ca2+ signaling to induce angiogenesis in cardiovascular disorders and, conversely, to alter tumor vasculature in cancer.

3. In addition to each Figure, the authors need to provide a Table that generally overview the pro-angiogenic and anti-angiogenic pathways which are associated with the regulation of endothelial Ca2+ signaling.

We thank the Reviewer for this observation. We added Table 1 to summarize the most relevant studies focusing on the pro-angiogenic endothelial Ca2+ toolkit. In particular, we described in the Table only the investigations assessing the involvement of more than one Ca2+ entry/ release pathways in the case of growth factors and the only one investigation addressing the Ca2+-related anti-angiogenic effect of angiostatin and endostatin.

4. In Figure 1, the red circle should be labeled Ca2+. In 2.1.2.1 ~ 2.1.2.3, it would be better to describe in more detail the mechanisms by which Ca2+ is released by InsP3R, RyR and TPC.

We thank the Reviewer for this observation. We labeled the red circle as Ca2+. As to Figure 2 - Figure 4, we would like to explain to the referee the rationale for drawing them: Figure 1 and Figure 5 were devoted to illustrate the basic mechanisms of endothelial Ca2+ signaling, while the remaining ones served to understand the Ca2+-dependency of a number of pro-angiogenic effectors. It is implicit that the Ca2+ entry/release pathways responsible for the Ca2+ response to VEGF refer to Figure 1. We specified this in their legends. We would like not change these Figures to maintain the focus on the Ca2+-dependent effectors, if the Referee agrees with our proposal.

We are grateful for your insightful revision and hope that you will not regard the manuscript worth being published on this Special Issue of the International Journal of Molecular Sciences.

With my kindest regards,

Francesco Moccia

Francesco Moccia, PhD

Laboratory of General Physiology,

Department of Biology and Biotechnology “L. Spallanzani”

University of Pavia,

Via Forlanini 6, 27100, Pavia, Italy.

Tel: 0039 0382 987614.

Fax: 0039 0382 987527.

Round 2

Reviewer 2 Report

This revised manuscript is acceptable for publication.